# Shallow neural networks trained to detect collisions recover features of visual loom-selective neurons

**Baohua Zhou[1,2], Zifan Li[2†], Sunnie Kim[2‡], John Lafferty[2,3]\*, Damon A Clark[1,4,5,6]\***

[1]Department of Molecular, Cellular and Developmental Biology, Yale University, New Haven, United States; [2]Department of Statistics and Data Science, Yale University, New Haven, United States; [3]Wu Tsai Institute, Yale University, New Haven, United States; [4]Interdepartmental Neuroscience Program, Yale University, New Haven, United States; [5]Department of Physics, Yale University, New Haven, United States; [6]Department of Neuroscience, Yale University, New Haven, United States

**\*For correspondence:**
john.lafferty@yale.edu (JL);
damon.clark@yale.edu (DAC)

**Present address:** [†]Facebook, Menlo Park, United States; [‡]Department of Computer Science, Princeton University, Princeton, United States

**Competing interest:** The authors declare that no competing interests exist.

**Abstract** Animals have evolved sophisticated visual circuits to solve a vital inference problem: detecting whether or not a visual signal corresponds to an object on a collision course. Such events are detected by specific circuits sensitive to visual looming, or objects increasing in size. Various computational models have been developed for these circuits, but how the collision-detection inference problem itself shapes the computational structures of these circuits remains unknown. Here, inspired by the distinctive structures of LPLC2 neurons in the visual system of *Drosophila*, we build anatomically-constrained shallow neural network models and train them to identify visual signals that correspond to impending collisions. Surprisingly, the optimization arrives at two distinct, opposing solutions, only one of which matches the actual dendritic weighting of LPLC2 neurons. Both solutions can solve the inference problem with high accuracy when the population size is large enough. The LPLC2-like solutions reproduces experimentally observed LPLC2 neuron responses for many stimuli, and reproduces canonical tuning of loom sensitive neurons, even though the models are never trained on neural data. Thus, LPLC2 neuron properties and tuning are predicted by optimizing an anatomically-constrained neural network to detect impending collisions. More generally, these results illustrate how optimizing inference tasks that are important for an animal's perceptual goals can reveal and explain computational properties of specific sensory neurons.

## Editor's evaluation

This paper trains a simple neural network model to perform a behaviorally important task: the detection of looming objects. Two solutions emerge, one of which shares several properties with the actual circuit. This is a nice demonstration that training a CNN on a behaviorally-relevant task can reveal how the underlying computations work.

## Introduction

For animals living in dynamic visual environments, it is important to detect the approach of predators or other dangerous objects. Many species, from insects to humans, rely on a range of visual cues to identify approaching, or looming, objects (*Regan and Beverley, 1978*; *Sun and Frost, 1998*; *Gabbiani et al., 1999*; *Card and Dickinson, 2008*; *Münch et al., 2009*; *Temizer et al., 2015*). Looming objects create characteristic visual flow fields. When an object is on a straight-line collision course with an animal, its edges will appear to the observer to expand radially outward, gradually occupying a larger

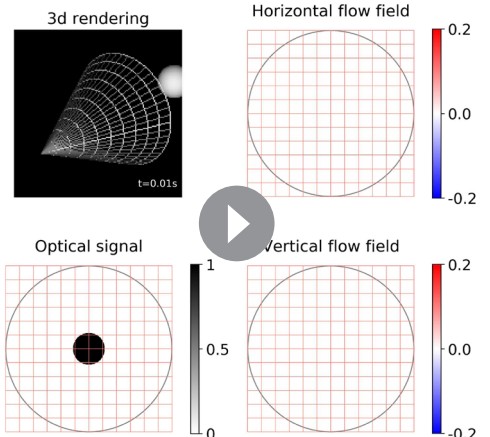

**Video 1.** Movie for a hit stimulus (single unit). Top left panel: 3d rendering as in the top row of Figure 3; bottom left panel: optical signal as in the second row of Figure 3; top right panel: flow fields in the horizontal direction as in rows 7 and 8 of Figure 3; bottom right panel: flow fields in the vertical direction as in rows 5 and 6 of Figure 3. Since we combined left (down) and right (up) flow fields in one panel, we used blue and red colors to indicate left (down) and right (up) directions, respectively. The movie has been slowed down by a factor of 5. All the movies shown in this paper can be found here: https://github.com/ClarkLabCode/LoomDetectionANN/tree/main/results/movies_exp.
https://elifesciences.org/articles/72067/figures#video1

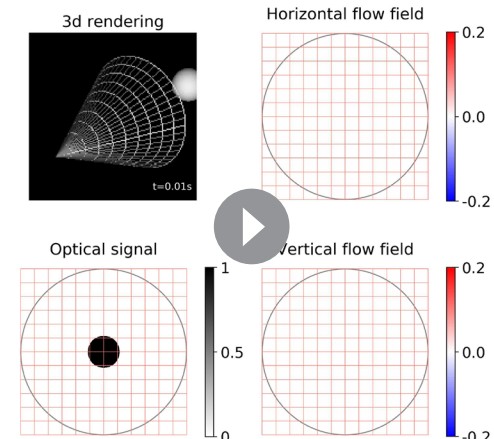

**Video 2.** Movie for a miss stimulus (single unit). The same arrangement as Video 1.
https://elifesciences.org/articles/72067/figures#video2

and larger portion of the visual field (*Video 1*). An object heading toward the animal, but which will not collide with it, also expands to occupy an increasing portion of the visual field, but its edges do not expand radially outwards with respect to the observer. Instead, they expand with respect to the object's center so that opposite edges can move in the same direction across the retina (*Video 2*). A collision detector must distinguish between these two cases, while also avoiding predicting collisions in response to a myriad of other visual flow fields, including those created by an object moving away (*Video 3*) or by the animal's own motion (*Video 4*). Thus, loom detection can be framed as a visual inference problem.

Many sighted animals solve this inference problem with high precision, thanks to robust loom-selective neural circuits evolved over hundreds of millions of years. The neuronal mechanisms for response to looming stimuli have been studied in a wide range of vertebrates, from cats and mice to zebrafish, as well as in humans (*King et al., 1992*; *Hervais-Adelman et al., 2015*; *Ball and Tronick,*

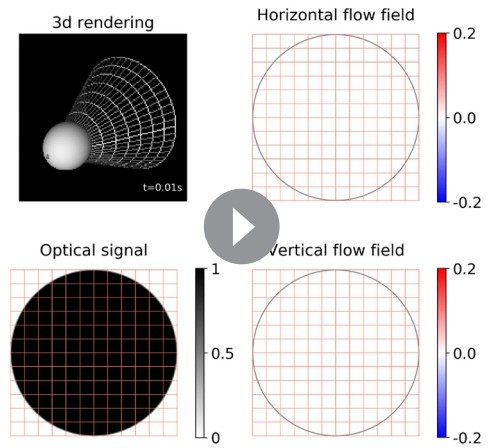

**Video 3.** Movie for a retreat stimulus (single unit). The same arrangement as Video 1.
https://elifesciences.org/articles/72067/figures#video3

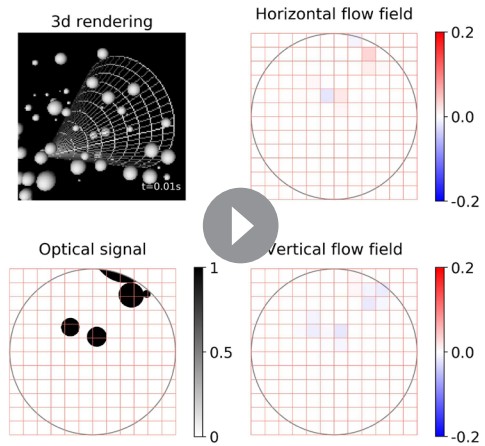

**Video 4.** Movie for a rotation stimulus (single unit). The same arrangement as Video 1.
https://elifesciences.org/articles/72067/figures#video4

*1971*; *Liu et al., 2011*; *Salay et al., 2018*; *Liu et al., 2011*; *Shang et al., 2015*; *Wu et al., 2005*; *Temizer et al., 2015*; *Dunn et al., 2016*; *Bhattacharyya et al., 2017*). In invertebrates, detailed anatomical, neurophysiological, behavioral, and modeling studies have investigated loom detection, especially in locusts and flies (*Oliva and Tomsic, 2014*; *Sato and Yamawaki, 2014*; *Santer et al., 2005*; *Rind and Bramwell, 1996*; *Card and Dickinson, 2008*; *de Vries and Clandinin, 2012*; *Muijres et al., 2014*; *Klapoetke et al., 2017*; *von Reyn et al., 2017*; *Ache et al., 2019*). An influential mathematical model of loom detection was derived by studying the responses of the giant descending neurons of locusts (*Gabbiani et al., 1999*). This model established a relationship between the timing of the neurons' peak responses and an angular size threshold for the looming object. Similar models have been applied to analyze neuronal responses to looming signals in flies, where genetic tools make it possible to precisely dissect neural circuits, revealing various neuron types that are sensitive to looming signals (*von Reyn et al., 2017*; *Ache et al., 2019*; *Morimoto et al., 2020*).

However, these computational studies did not directly investigate the relationship between the structure of the loom-sensitive neural circuits and the inference problem they appear to solve. On the one hand, the properties of many sensory circuits appear specifically tuned to the tasks that they are executing (*Turner et al., 2019*). In particular, by taking into account relevant behaviors mediated

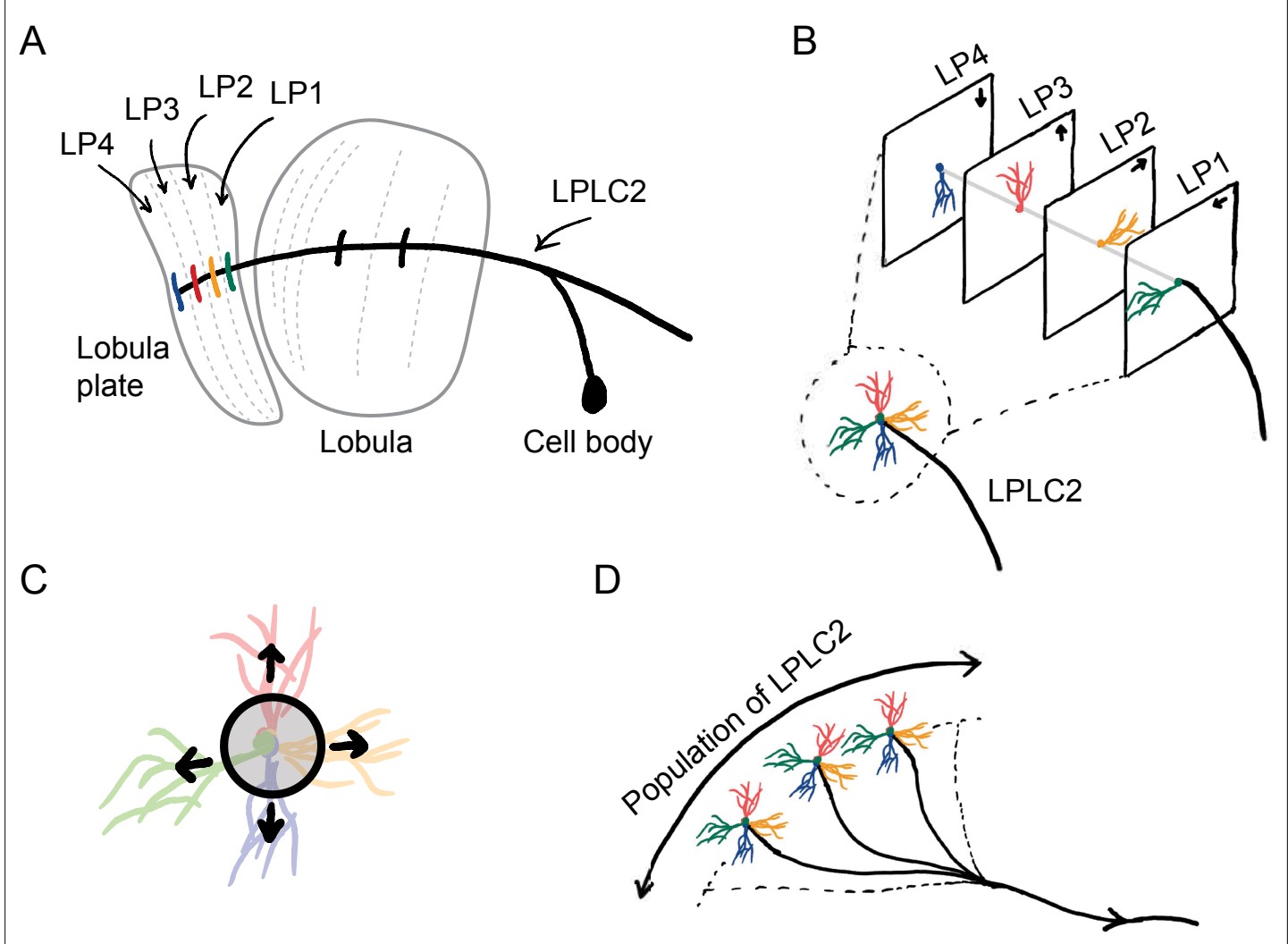

**Figure 1.** Sketches of the anatomy of LPLC2 neurons (*Klapoetke et al., 2017*). (**A**) An LPLC2 neuron has dendrites in lobula and the four layers of the lobula plate (LP): LP1, LP2, LP3, and LP4. (**B**) Schematic of the four branches of the LPLC2 dendrites in the four layers of the LP. The arrows indicate the preferred direction of motion sensing neurons with axons in each LP layer (*Maisak et al., 2013*). (**C**) The outward dendritic structure of an LPLC2 neuron is selective for the outwardly expanding edges of a looming object (black circle). (**D**) The axons of a population of more than 200 LPLC2 neurons converge to the giant fibers, descending neurons that mediate escape behaviors (*Ache et al., 2019*).

by specific sensory neurons, experiments can provide insight into their tuning properties (*Krapp and Hengstenberg, 1996*; *Sabbah et al., 2017*). On the other hand, computational studies that have trained artificial neural networks to solve specific visual and cognitive tasks, such as object recognition or motion estimation, have revealed response patterns similar to the corresponding biological circuits (*Yamins et al., 2014*; *Yamins and DiCarlo, 2016*; *Richards et al., 2019*) or even individual neurons (*Mano et al., 2021*). Thus, here we ask whether we can reproduce the properties associated with neural loom detection simply by optimizing shallow neural networks for collision detection.

The starting point for our computational model of loom detection is the known neuroanatomy of the visual system of the fly. In particular, the loom-sensitive neuron LPLC2 (lobula plate/lobula columnar, type 2) has been studied in detail (*Wu et al., 2016*). These neurons tile visual space, sending their axons to descending neurons called the giant fibers (GFs), which trigger the fly's jumping and take-off behaviors (*Tanouye and Wyman, 1980*; *Card and Dickinson, 2008*; *von Reyn et al., 2017*; *Ache et al., 2019*). Each LPLC2 neuron has four dendritic branches that receive inputs at the four layers of the lobula plate (LP) (*Figure 1A*; *Maisak et al., 2013*; *Klapoetke et al., 2017*). The retinotopic LP layers host the axon terminals of motion detection neurons, and each layer uniquely receives motion information in one of the four cardinal directions (*Maisak et al., 2013*). Moreover, the physical extensions of the LPLC2 dendrites align with the preferred motion directions in the corresponding LP layers (*Figure 1B*; *Klapoetke et al., 2017*). These dendrites form an outward radial structure, which matches the moving edges of a looming object that expands from the receptive field center (*Figure 1C*). Common stimuli such as the wide-field motion generated by movement of the insect only match part of the radial structure, and strong inhibition for inward-directed motion suppresses responses to such stimuli. Thus, the structure of the LPLC2 dendrites favors responses to visual stimuli with edges moving radially outwards, corresponding to objects approaching the receptive field center.

The focus of this paper is to investigate how loom detection in LPLC2 can be seen as the solution to a computational inference problem. Can the structure of the LPLC2 neurons be explained in terms of optimization—carried out during the course of evolution—for the task of predicting which trajectories will result in collisions? How does coordination among the population of more than 200 LPLC2

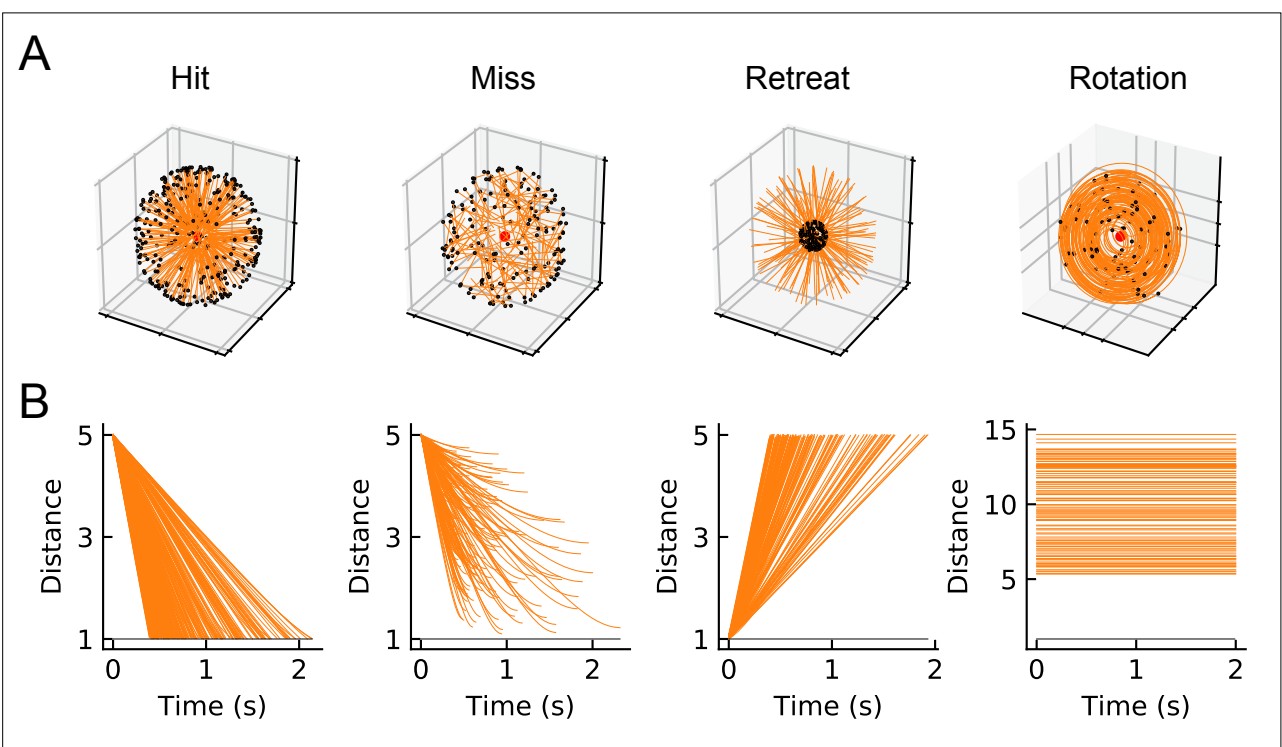

**Figure 2.** Four types of synthetic stimuli (Materials and methods). (**A**) Orange lines represent trajectories of the stimuli. The black dots represent the starting points of the trajectories. For hit, miss, and retreat cases, multiple trajectories are shown. For rotation, only one trajectory is shown. (**B**) Distances of the objects to the fly eye as a function of time. Among misses, only the approaching portion of the trajectory was used. The horizontal black lines indicate the distance of 1, below which the object would collide with the origin.

neurons tiling a fly's visual system affect this optimization? To answer these questions, we built simple anatomically-constrained neural network models, which receive motion signals in the four cardinal directions. We trained the model using artificial stimuli to detect visual objects on a collision course with the observer. Surprisingly, optimization finds two distinct types of solutions, with one resembling the LPLC2 neurons and the other having a very different configuration. We analyzed how each of these solutions detects looming events and where they show distinct individual and population behaviors. When tested on visual stimuli not in the training data, the optimized solutions with filters that resemble LPLC2 neurons exhibit response curves that are similar to those of LPLC2 neurons measured experimentally (*Klapoetke et al., 2017*). Importantly, although it only receives motion signals, the optimized model shows characteristics of an angular size encoder, which is consistent with many biological loom detectors, including LPLC2 (*Gabbiani et al., 1999*; *von Reyn et al., 2017*; *Ache et al., 2019*). Our results show that optimizing a neural network to detect looming events can give rise to the properties and tuning of LPLC2 neurons.

## Results
### A set of artificial visual stimuli is designed for training models
Our goal is to compare computational models trained to perform loom-detection with the biological computations in LPLC2 neurons. We first created a set of stimuli to act as training data for the inference task (Materials and methods). We considered the following four types of motion stimuli: loom-and-hit (abbreviated as hit), loom-and-miss (miss), retreat, and rotation (*Figure 2*). The hit stimuli consist of a sphere that moves in a straight line towards the origin on a collision course (*Figure 3*, *Video 1*). The miss stimuli consist of a sphere that moves in a straight line toward the origin but misses it (*Figure 3*, *Video 2*). The retreat stimuli consist of a sphere moving in a straight line away from the origin (*Figure 3*, *Video 3*). The rotation stimuli consist of objects rotating about an axis that goes through the origin (*Figure 3*, *Video 4*). All stimuli were designed to be isotropic; the first three stimuli could have any orientation in space, while the rotation could be about any axis through the origin (*Figure 2*). All trajectories were simulated in the frame of reference of the fly at the origin, with distances measured with respect to the origin. For simplicity, the fly is assumed to be a point particle with no volume (Red dots in *Figure 2* and the apexes of the cones in *Figure 3*). For hit, miss, and retreat stimuli, the spherical object has unit radius, and for the case of rotation, there were 100 objects of various radii scattered isotropically around the fly (*Figure 3*).

### An anatomically constrained mathematical model
We designed and trained simple, anatomically constrained neural networks (*Figure 4*) to infer whether or not a moving object will collide with the fly. The features of these networks were designed to mirror anatomical features of the fly's LPLC2 neurons (*Figure 1*). We will consider two types of single units in our models (*Figure 4*): a linear receptive field (LRF) unit and a rectified inhibition (RI) unit. Both types of model units receive input from a 60 degree diameter cone of visual space, represented by white cones and grey circles in *Figure 3*, approximately the same size as the receptive fields measured in LPLC2 (*Klapoetke et al., 2017*). The four stimulus sets were projected into this receptive field for training and evaluating the models. The inputs to the units are local directional signals computed in the four cardinal directions at each point of the visual space: downward, upward, rightward, and leftward (*Figure 3*). These represent the combined motion signals from T4 and T5 neurons in the four layers of the lobula plate (*Maisak et al., 2013*). They are computed as the non-negative components of a Hassenstein-Reichardt correlator model (*Hassenstein and Reichardt, 1956*) in both horizontal and vertical directions (*Figure 3—figure supplement 1*; Materials and methods). The motion signals are computed with a spacing of 5 degrees, roughly matching the spacing of the ommatidia and processing columns in the fly eye (*Stavenga, 2003*).

For the LRF model unit (*Figure 4A*), there is a single set of four real-valued filters, the elements of which can be positive (excitatory, red) or negative (inhibitory, blue). The four filters integrate motion signals from the four cardinal directions, respectively, and their outputs are summed and rectified to generate the output of a single model unit (*Figure 4A*). These spatial filters effectively represent excitatory inputs to LPLC2 directly from T4 and T5 in the LP (positive elements), and inhibitory inputs mediated by local interneurons (negative elements) (*Mauss et al., 2015*; *Klapoetke et al., 2017*). All

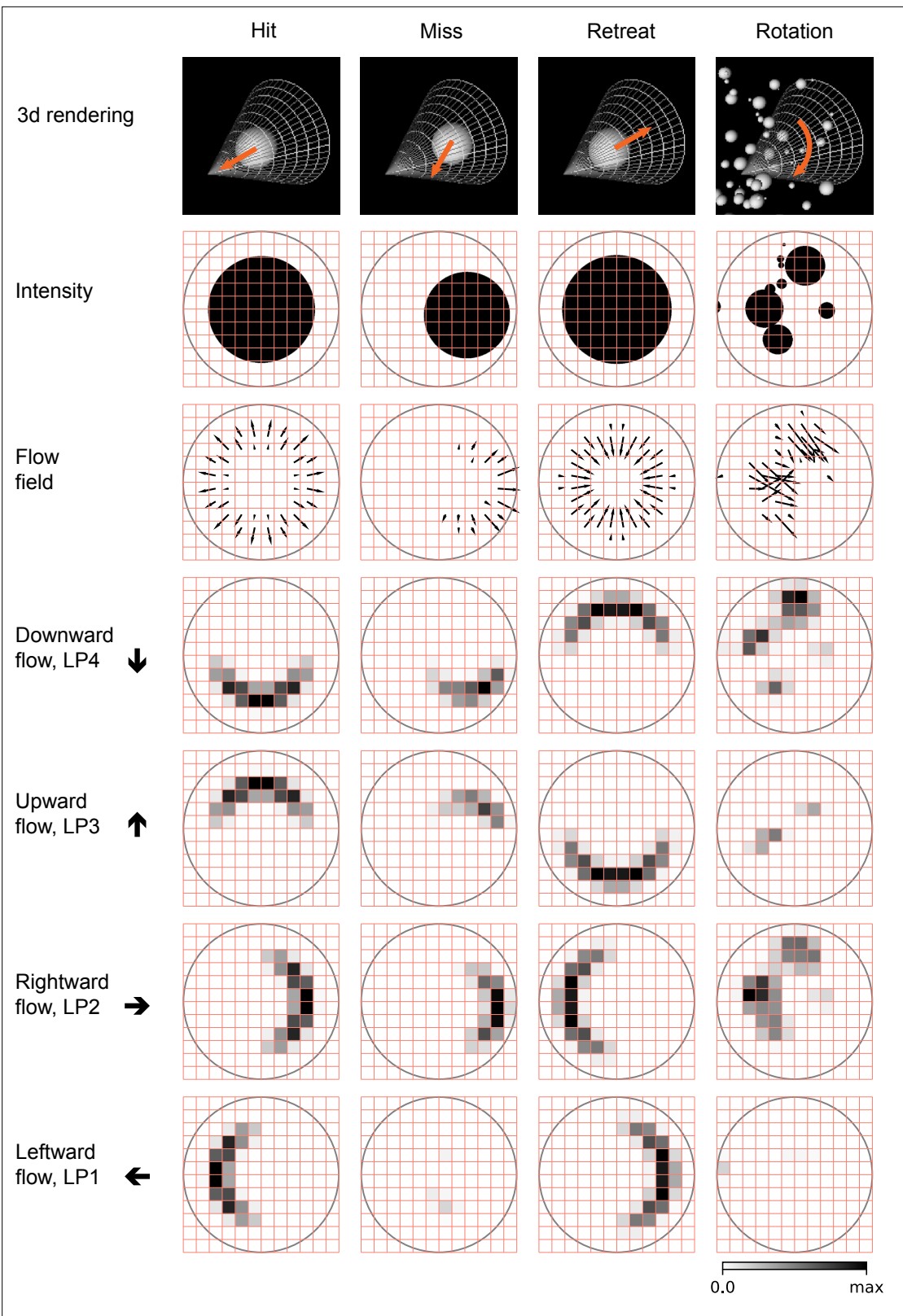

**Figure 3.** Snapshots of optical flows and flow fields calculated by a Hassenstein Reichardt correlator (HRC) model (*Figure 3—figure supplement 1*, Materials and methods) for the four types of stimuli (*Figure 2*). First row: 3d rendering of the spherical objects and the LPLC2 receptive field (represented by a cone) at a specific time in the trajectory. The orange arrows indicate the motion direction of each object. Second row: 2d projections of the objects (black shading) within the LPLC2 receptive field (the gray circle). Third row: the thin black arrows indicate flow fields generated by the

*Figure 3 continued on next page*

*Figure 3 continued*

edges of the moving objects. Forth to seventh rows: decomposition of the flow fields in the four cardinal directions with respect to the LPLC2 neuron under consideration: downward, upward, rightward, and leftward, as indicated by the thick black arrows. These act as models of the motion signal fields in each layer of the LP.

The online version of this article includes the following figure supplement(s) for figure 3:

**Figure supplement 1.** Tuning curve of HRC motion estimator and distributions of the estimated flow fields.

filters act on the 60 degree receptive field of a unit. A 90 degree rotational symmetry is imposed on the filters, so that the filters in each layer are identical. Moreover, each filter is symmetric about the axis of motion (Materials and methods). Although these symmetry assumptions are not necessary and may be learned from training (*Figure 5—figure supplement 3*), they greatly reduce the number of parameters in the models. No further assumptions were made about the structure of the filters. Note that the opposing spatial patterns of the excitatory and inhibitory components in *Figure 4* are only for illustration purposes, and are not imposed on the models. The LRF model unit is equivalent to a linear-nonlinear model and constitutes one of the simplest possible models for the LPLC2 neurons. In this work, we will focus most of our analysis on this simplest form of the model.

In addition, we will also consider a more complex model unit, which we call the rectified inhibition (RI) model unit (*Figure 4B*). In this unit, all the same filter symmetries are enforced, but there are two sets of non-negative filters: a set of excitatory filters and a set of inhibitory filters. The RI unit incorporates a fundamental difference between the excitatory and inhibitory filters: while the integrated signals from each excitatory filter are sent directly to the downstream computations, the integrated signals from each inhibitory filter are rectified before being sent downstream. The outputs of the eight filters are summed and rectified to generate the output of a single model unit in response to a given stimulus (*Figure 4B*). If one removes the rectification of the inhibitory filters (as is possible with an appropriate choice of parameters), the RI unit becomes equivalent to the LRF unit (Materials and methods). This difference between the two model units reflects different potential constraints on the inhibitory inputs to an actual LPLC2 neuron. While the excitatory inputs to LPLC2 are direct connections, the inhibitory inputs are mediated by inhibitory interneurons (LPi) between LP layers (*Mauss et al., 2015*; *Klapoetke et al., 2017*). The LRF model unit assumes that LPi interneurons are linear and do not strongly rectify signals, while the RI model unit approximates rectified transmission by LPi interneurons.

In the fly brain, a population of LPLC2 neurons converges onto the GFs (*Figure 1D*). Accordingly, in our model there are $M$ replicates of model units, with orientations that are spread uniformly over the $4\pi$ steradians of the unit sphere (*Figure 4C and D*, *Figure 4—figure supplement 1*, Materials and methods). In this way, the receptive fields of the $M$ units roughly tile the whole angular space, with or without overlap, depending on the value of $M$. The sum of the responses of the $M$ model units is fed into a sigmoid function to generate the predicted probability of collision for a given trajectory (Materials and methods). The loss function then is defined as the cross entropy between the predicted probabilities and the stimulus labels (hits are labeled one and all others are labeled 0).

## Optimization finds two distinct solutions to the loom-inference problem

The objective of this study is to investigate how the binary classification task shapes the structure of the filters, and how the number of units $M$ affects the results. We begin with the simplest LRF model, which possesses only a single unit, $M = 1$. After training with 200 random initializations of the filters, we find that the converged solutions fall into three broad categories (*Figure 5*, *Figure 5—figure supplement 1*). Two solutions have spatial structures that are—surprisingly—roughly opposite from one another (magenta and green). Based on the configurations of the positive-valued elements (stronger excitation) of the filters (Materials and methods), we call one solution type *outward solutions* (magenta) and the other type *inward solutions* (green) (*Figure 5C*, *Figure 5—figure supplement 1*). In this single-unit model, the inward solutions have higher area under the curve (AUC) scores for both receiver operating characteristic (ROC) and precision recall curves, and thus perform better than the outward solutions on the discrimination task (*Figure 5D*). A third category of solution has all the elements in the filters very close to zero (*zero solutions* in black squares, *Figure 5—figure supplement 1*). This solution appears roughly 5–15% of the time, and appears to be a local minimum of

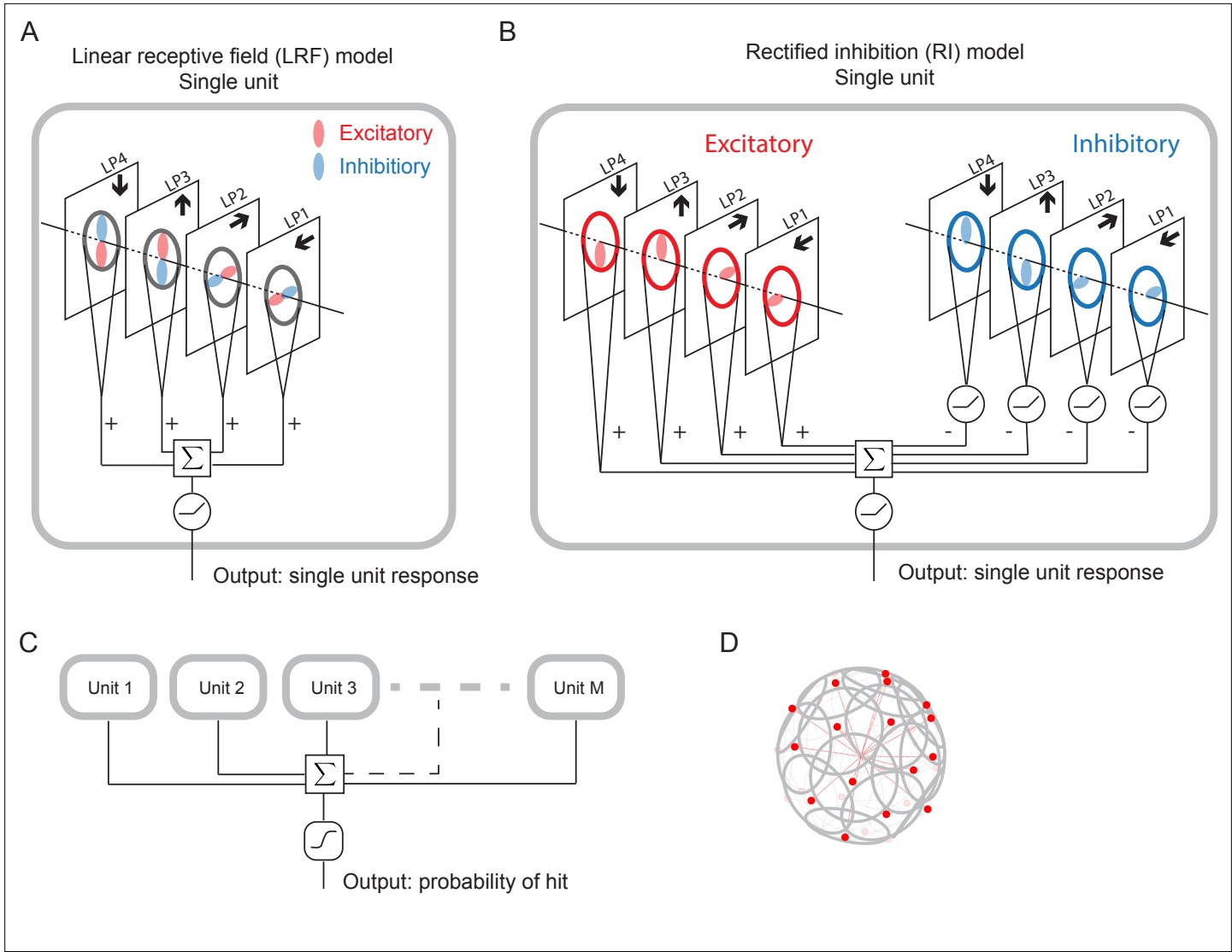

**Figure 4.** Schematic of the models (Materials and methods). (**A**) Single LRF model unit. There are four linear spatial filters, labeled LP4, LP3, LP2, and LP1, which correspond to the four LP layers (*Figure 1*). Each filter has real-valued elements, and if the element is positive (negative), it is excitatory (inhibitory), represented by the color red (blue). The opposing spatial arrangement of the excitatory and inhibitory filters are illustrative, and do not represent constraints on the model. Each filter receives a field of motion signals from the corresponding layer of the model LP (fourth to seventh rows in *Figure 3*), indicated by the four black arrows (*Figure 1*). The four filtered signals are summed together before a rectifier is applied to produce the output, which is the response of a single unit. (**B**) Single RI model unit. There are two sets of nonnegative filters: excitatory (red) and inhibitory (blue). Each set has four filters, and each filter receives the same motion signals as the corresponding one in the LRF unit. The weighted signals from the excitatory filters and the inhibitory filters (rectified) are pooled together before a rectifier is applied to produce the output, which is the response of a single unit. When the inhibitory filters are not rectified, this model effectively reduces to the LRF model in (**A**) (Materials and methods). (**C**) The outputs from $M$ units are summed and fed into a sigmoid function to estimate the probability of hit. (**D**) The $M$ units have their orientations almost evenly distributed in angular space. Red dots represent the centers of the receptive fields and the grey lines represent the boundaries of the receptive fields on unit sphere. The red lines are drawn from the origin to the center of each receptive field.

The online version of this article includes the following figure supplement(s) for figure 4:

**Figure supplement 1.** Coordinate system for model and stimuli.

the optimization, dependent on the random initialization. This uninteresting category of solutions is ignored in subsequent analyses.

As the number of units $M$ increases, the population of units covers more angular space, and when $M$ is large enough ($M \geq 16$), the receptive fields of the units begin to overlap with one another (*Figure 6A*). In the fly visual system there are over 200 LPLC2 neurons across both eyes (*Ache et al.,*

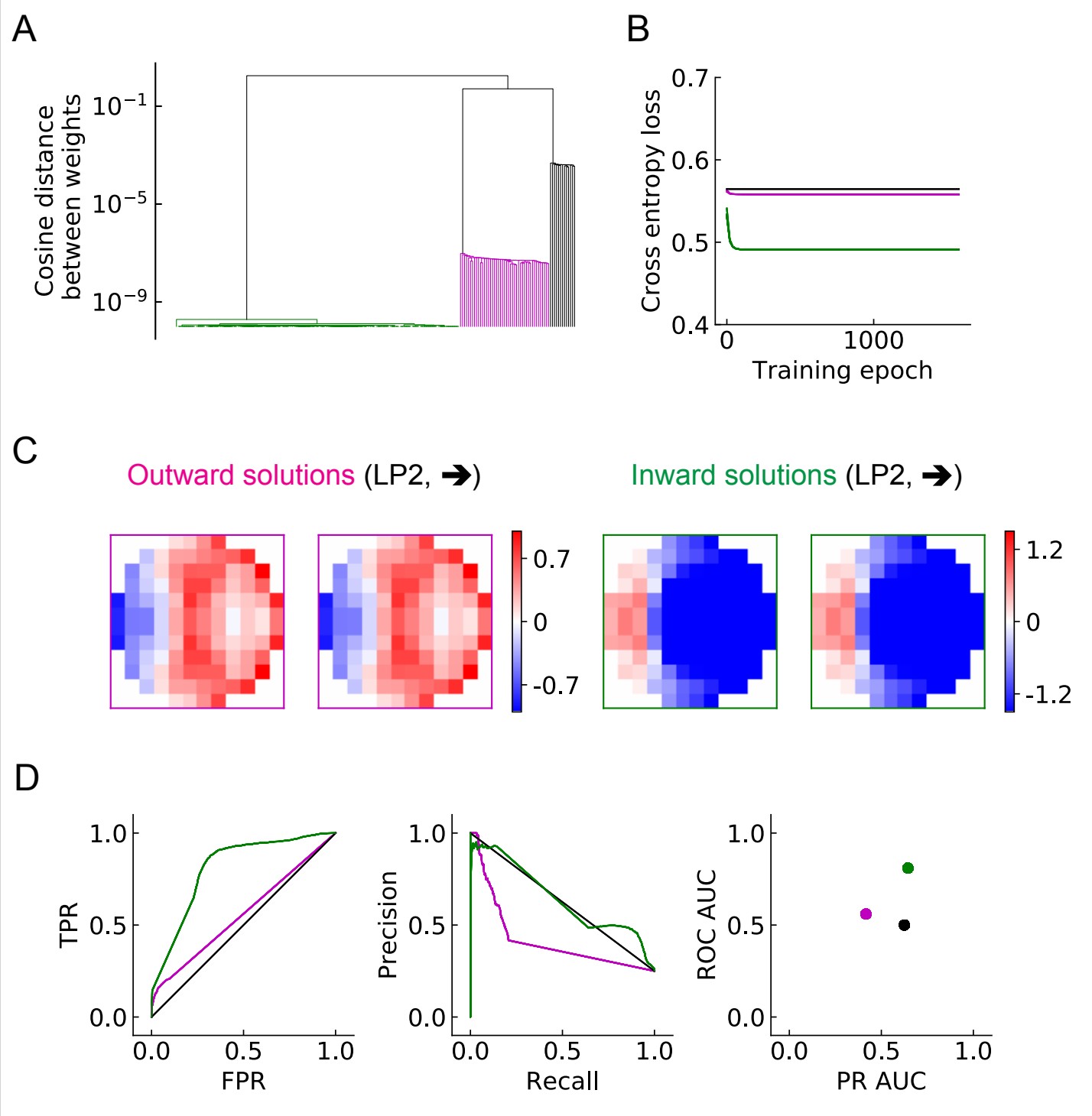

**Figure 5.** Three distinct types of solutions appear from training a single unit on the binary classification task (LRF model). (**A**) Clustering of the trained filters/weights shown as a dendrogram (Materials and methods). Different colors indicate different clusters, which are preserved for the rest of the paper: outward, inward, and zero solutions are magenta, green, and black, respectively. (**B**) The trajectories of the loss functions during training. More than one example are shown for each type of solution, but lines fall on top of one another. (**C**) Two distinct types of solutions are represented by two types of filters that have roughly opposing structures: an outward solution (magenta boxes) and an inward solution (green boxes). For each solution type, two examples of the trained filters from different initializations are shown; they are almost identical. The third type of solution (color black in (**A**) and (**B**)) has filter elements all close to zero. We call these zero solutions (*Figure 5—figure supplement 1*). (**D**) Performance of the three solution types (Materials and methods). TPR: true positive rate; FPR: false positive rate; ROC: receiver operating characteristic; PR: precision recall; AUC: area under the curve. More than one example is shown for each type of solution, but lines and dots with the same color fall on top of one another.

*Figure 5 continued on next page*

*Figure 5 continued*

The online version of this article includes the following figure supplement(s) for figure 5:

**Figure supplement 1.** More examples of the trained filters for the three types of solutions.

**Figure supplement 2.** As in the main figure but for the RI model.

**Figure supplement 3.** Examples of the trained outward and inward filters without imposed symmetries.

*2019*), which corresponds to a dense distribution of units. This is illustrated by the third row in *Figure 6A*, where $M = 256$. When $M$ is large, objects approaching from any direction are detectable, and such object signals can be detected simultaneously by many neighboring units. The two oppositely structured solutions persist, regardless of the value of $M$ (*Figure 6*, *Figure 6—figure supplement 1*, *Figure 6—figure supplement 2*, *Figure 6—figure supplement 3*). Strikingly, the inhibitory component of the outward solutions becomes broader as $M$ increases and expands to extend across the entire receptive field (*Figure 6B*). This broad inhibition is consistent with the large receptive field of LPi neurons suggested by experiments (*Mauss et al., 2015*; *Klapoetke et al., 2017*). The outward, inward, and zero solutions all also appear in trained solutions of the RI models. (*Figure 5—figure supplement 2*, *Figure 6—figure supplement 3*).

Units with outward-oriented filters are activated by motion radiating outwards from the center of the receptive field, such as the hit event illustrated in *Figure 3*. These excitatory components resemble the dendritic structures of the actual LPLC2 neurons observed in experiments, where for example, the rightward motion-sensitive component (LP2) occupies mainly the right side of the receptive field. In the outward solutions of the LRF models, the rightward motion-sensitive inhibitory components mainly occupy the *left* side of the receptive field (*Figure 6B*, *Figure 6—figure supplement 2*). This is also consistent with the properties of the lobula plate intrinsic (LPi) interneurons, which project inhibitory signals roughly retinotopically from one LP layer to the adjacent layer with opposite directional tuning (*Mauss et al., 2015*; *Klapoetke et al., 2017*).

The unexpected inward-oriented filters have the opposite structure. In the inward solutions, the rightward sensitive excitatory component occupies the left side of the receptive field, and the inhibitory component occupies the right side. Such weightings make the model selective for motion converging toward the receptive field center, such as the retreat event shown in *Figure 3*. This is a puzzling structure for a loom detector, and warrants a more detailed exploration of the response properties of the inward and outward solutions.

## Units with outward and inward filters respond to hits originating in distinct regions

To understand the differences between the two types of solutions and why the inward ones can predict collisions, we investigated how units respond to hit stimuli originating at different angles $\theta$ (*Figure 7A*). When there is no signal, the baseline activity of outward units is zero; however, the baseline activity of inward units is above zero (grey dashed lines in *Figure 7B and C*). This is because the trained intercepts are negative (positive) in the outward (inward) case and when the input is zero (no signal), the unit activity cannot (can) get through the rectifier (Materials and methods). (The training did not impose any requirements on these intercepts.) The outward units respond strongly to stimuli originating near the center of the receptive field, but do not respond to stimuli originating at angles larger than approximately $30°$ (*Figure 7B and C*). In contrast, inward units respond below baseline to hit stimuli approaching from the center and above baseline to stimuli approaching from the periphery of the receptive field, with $\theta$ between roughly $30°$ and $90°$ (*Figure 7B and C*). This helps explain why the inward units can act as loom detectors: they are sensitive to hit stimuli coming from the edges of the receptive field rather than from the center. The hit stimuli are isotropic (*Figure 2A*), so the number of stimuli with angles between $30°$ and $90°$ is much larger than the number of stimuli with angles below $30°$ (*Figure 7D*). Thus, the inward units are sensitive to more hit cases than the outward ones. One may visualize these responses as heat maps of the mean response of the units in terms of object distance to the fly and the incoming angle (*Figure 7E*). For the hit cases, the response patterns are consistent with the intuition about trajectory angles (*Figure 7C*). Both outward and inward units respond less strongly to miss signals than to hit signals. As expected, while the outward units respond at most weakly to retreating signals, the inward ones respond to these signals with angles near $180°$, since the motion

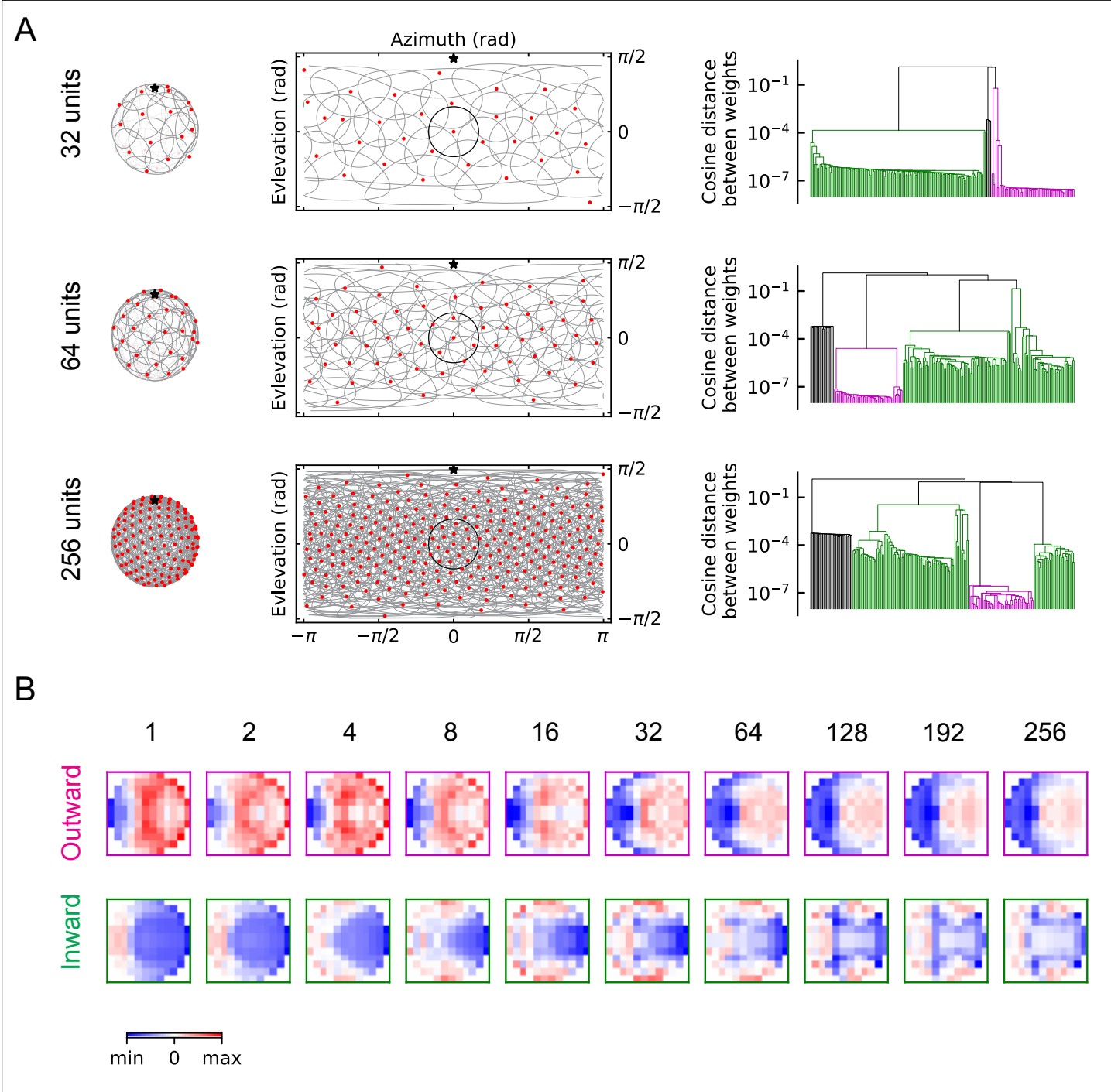

**Figure 6.** The outward and inward solutions also arise for models with multiple units (LRF models). (**A**) Left column: angular distribution of the units, where red dots are centers of the receptive fields, the grey circles are the boundaries of the receptive fields, and the black star indicates the top of the fly head. Middle column: 2d map of the units with the same symbols as in the left column, with one unit highlighted in black. Right column: clustering results shown as dendrogams with color codes as in *Figure 5*. (**B**) Examples of the trained filters for outward and inward solutions with different numbers of units.

The online version of this article includes the following figure supplement(s) for figure 6:

**Figure supplement 1.** Performance of the different solutions (LRF models).

**Figure supplement 2.** More examples of the outward and inward filters (LRF models).

**Figure supplement 3.** Examples of the outward and inward filters for RI models.

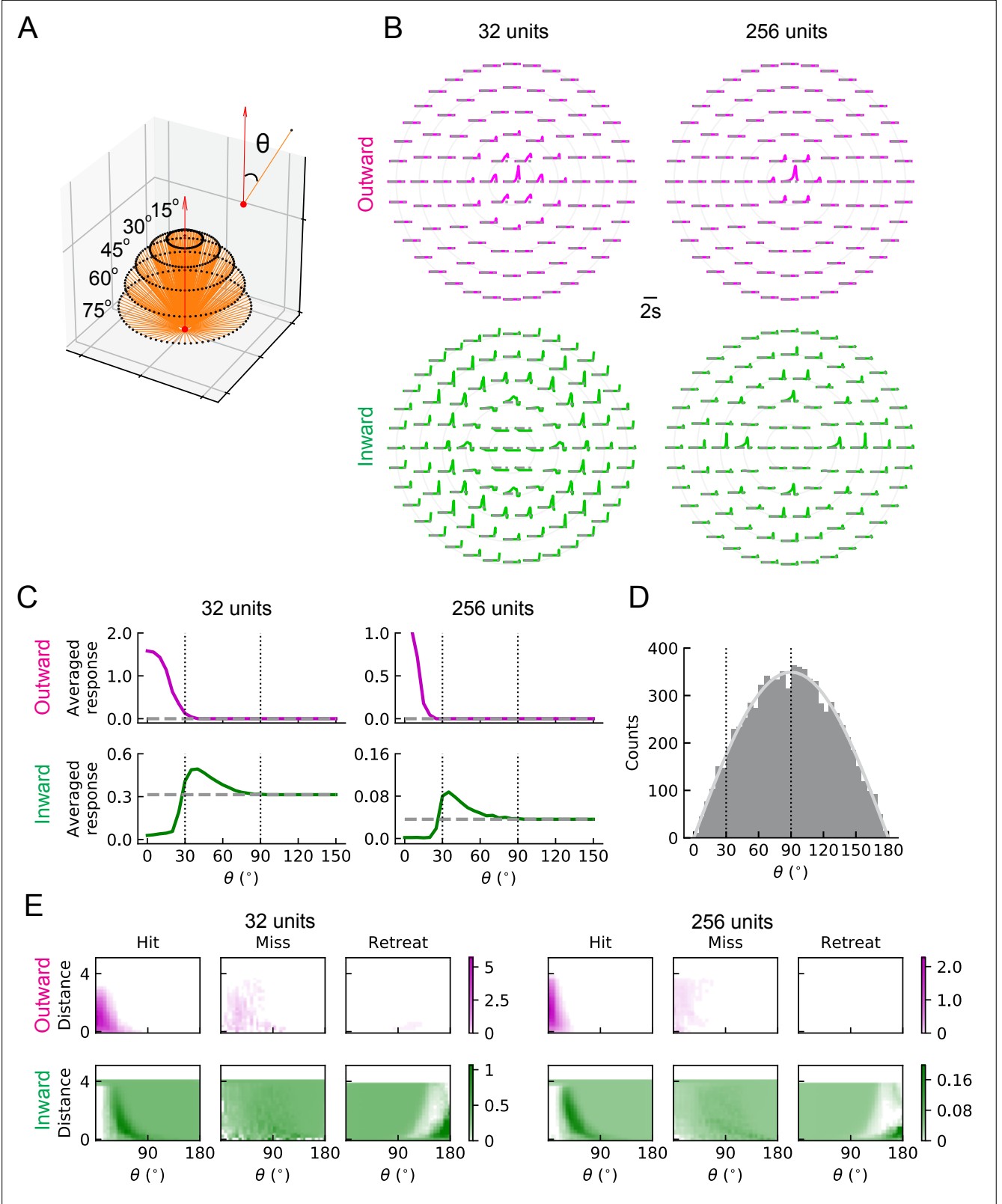

**Figure 7.** LRF units with outward and inward filters show distinct patterns of responses. (**A**) Trajectories of hit stimuli originating at different angles from the receptive field center, denoted by $\theta$. Symbols are the same as in *Figure 2* except that the upward red arrow represents the orientation of one unit (z direction, *Figure 4—figure supplement 1*). The numbers with degree units indicate the specific values of the incoming angles of different hit trajectories. (**B**) Response patterns of a single unit with either outward (magenta) or inward (green) filters obtained from optimized solutions with 32

*Figure 7 continued on next page*

*Figure 7 continued*

and 256 units, respectively. The horizontal gray dashed lines show the baseline activity of the unit when there is no stimulus. The solid grey concentric circles correspond to the values of the incoming angles in (**A**). The responses have been scaled so that each panel has the same maximum value. (**C**) Temporally averaged responses against the incoming angle $\theta$ in (**A**). Symbols and colors are as in (**B**). (**D**) Histogram of the incoming angles for the hit stimuli in *Figure 2A*. The gray curve represents a scaled sine function equal to the expected probability for isotropic stimuli. (**E**) Heatmaps of the response of a single unit against the incoming angle $\theta$ and the distance to the fly head, for both outward and inward filters obtained from optimized models with 32 and 256 units, respectively. The responses were calculated using the stimuli in *Figure 2*.

The online version of this article includes the following figure supplement(s) for figure 7:

**Figure supplement 1.** As in the main figure but for the RI units.

of edges in such cases is radially inward. The RI model units have similar response patterns to the LRF model units (*Figure 7—figure supplement 1*).

## Outward solutions have sparse codings and populations of units accurately predict hit probabilities

Individual units of the two solutions are very different from one another in both their filter structure and their response patterns to different stimuli. In populations of units, the outward and inward solutions also exhibit very different response patterns for a given hit stimulus (*Figure 8A and B*, *Videos 5 and 6*). In particular, active outward units usually respond more strongly than inward units, but more inward units will be activated by a hit stimulus. This is consistent with the findings above, in which inward filter shapes responded to hits arriving from a wider distribution of angles (*Figure 7*). For all the four types of stimuli, the outward solutions generally show relatively sparse activity among units, especially for models with larger numbers of units $M$, while the inward solutions show broader activity among units (*Figure 8A and B*, *Figure 8—figure supplement 1*, *Figure 8—figure supplement 2*, *Videos 5–12*).

When a population of units encodes stimuli, at each time point, the sum of the activities of the units is used to infer the probability of hit. In our trained models, the outward and inward solutions predict similar trajectories of probabilities of hit (*Figure 8A*). The outward solutions suppress the miss and retreat signals better, while the inward solutions better suppress rotation signals (*Figure 8C*). Both 32 unit and 256 unit models have units covering the entire visual field (*Figure 6*), but the models with 256 units can more accurately detect hit stimuli (*Figure 8C*). In some cases, misses can appear very similar to hits if the object passes near the origin. Both inward and outward solutions reflect this in their predictions in response to near misses, which have higher hit probabilities than far misses (*Figure 8D*).

The inward and outward solutions of RI models have similar behaviors to the LRF models (*Figure 8—figure supplement 3*). This indicates that the linear receptive field intuition captures behavior of the potentially more complicated RI model, as well.

## Large populations of units improve performance

Since a larger number of units will cover a larger region of the visual field, a larger population of units can in principle provide more information about the incoming signals. In general, the models perform better as the number of units $M$ increases (*Figure 9A*). When $M$ is above 32, both the ROC-AUC and PR-AUC scores are almost 1 (Materials and methods), which indicates that the model is very accurate on the binary classification task presented by the four types of synthetic stimuli. As $M$ increases, the outward solutions become closer to the inward solutions in terms of both AUC score and cross entropy loss (*Figure 9A and B*). This is also true for the RI models (*Figure 9—figure supplement 1A and B*).

Beyond performance of the two solution types, we also calculated the ratio of the number of outward to inward solutions in 200 random initializations of the models with $M$ units. For the LRF models, as the number of units increases, the ratio remained relatively constant, fluctuating around 0.5 (*Figure 9—figure supplement 2*). For the RI models, on the other hand, as the number of units increases, an increasing proportion of solutions have outward filters (*Figure 9—figure supplement 2*). For RI models with 256 units, the chance that an outward filter appears as a solution is almost 90% compared with roughly 50% when $M = 1$.

Because of this qualitative difference between the LRF and RI models, we next asked whether the form of the nonlinearity in the LRF model could influence the solutions found through optimization. When we replaced the rectified linear unit (ReLU) in LRF models with the exponential linear unit (ELU)

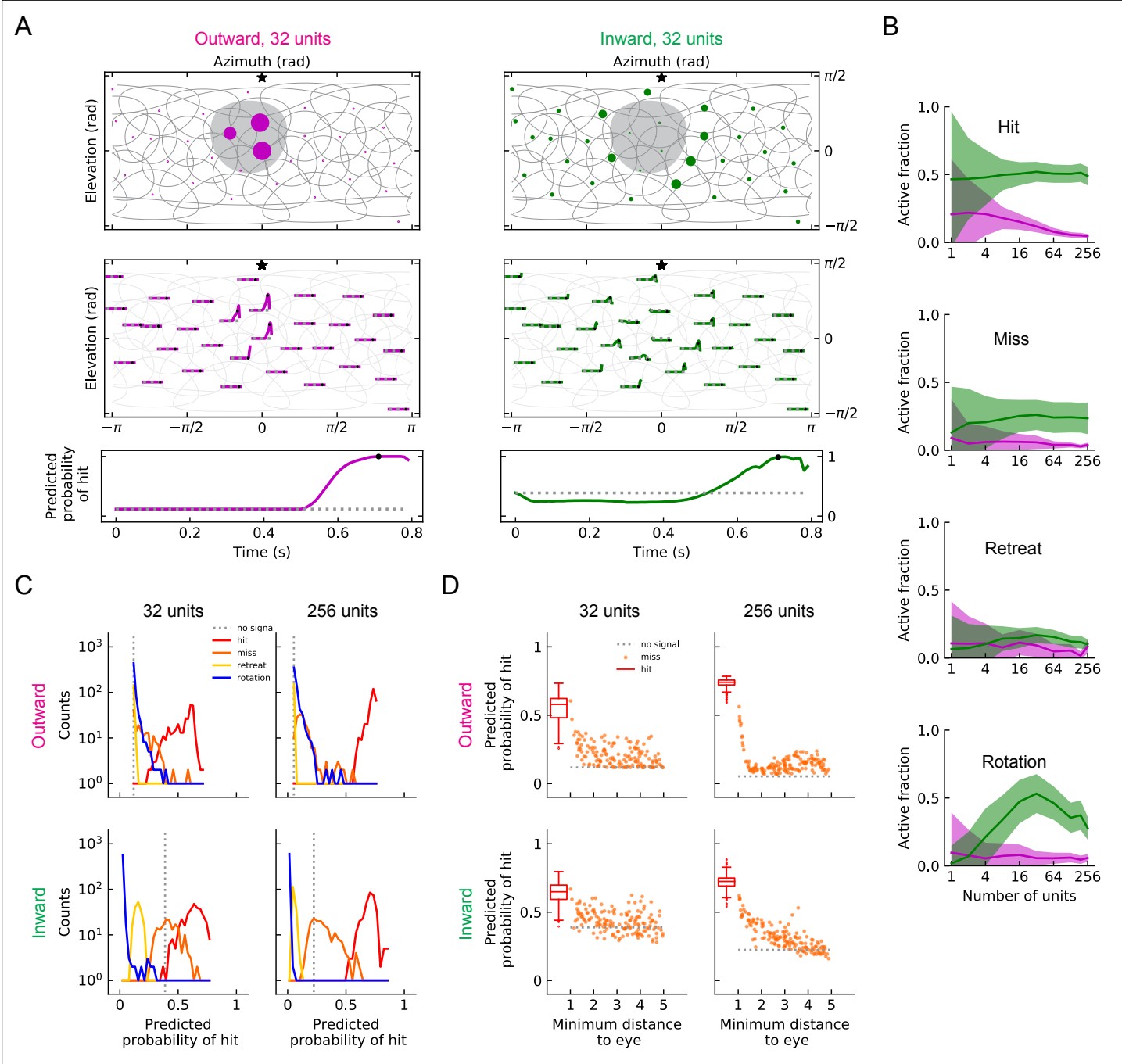

**Figure 8.** Population coding of stimuli (LRF models). (**A**) Top row: snapshots of the unit responses of outward solutions (magenta dots) and inward solutions (green dots) for a hit stimulus. The size of the dots represents the strength of the response. The gray shading represents the looming object in the snapshot. See also *Videos 5 and 6*. Symbols and colors are as in *Figure 6*. Middle row: time traces of the responses for the same hit stimulus as in the top row. Time proceeds to the right in each trace. Bottom row: time trace of the probability of hit for the same hit stimulus as in the top row (Materials and methods). Black dots in the middle and bottom rows indicate the time of the snapshot in the top row. The dotted gray line represents the basal model response. (**B**) Fractions of the units that are activated above the baseline by different types of stimuli (hit, miss, retreat, rotation) as a function of the number of units $M$ in the model. The lines represent the mean values averaged across stimuli, and the shaded areas show one standard deviation (Materials and methods). (**C**) Histograms of the probability of hit inferred by models with 32 or 256 units for the four types of synthetic stimuli (Materials and methods). (**D**) The inferred probability of hit as a function of the minimum distance of the object to the fly eye for the miss cases. For comparison, the hit distribution is represented by a box plot (the center line in the box: the median; the upper and lower boundaries of the box: 25% and 75% percentiles; the upper and lower whiskers: the minimum and maximum of non-outlier data points; the circles: outliers).

The online version of this article includes the following figure supplement(s) for figure 8:

*Figure 8 continued on next page*

*Figure 8 continued*

**Figure supplement 1.** Geometry of responses as in *Figure 8A*, but for miss and retreat stimuli (LRF models).

**Figure supplement 2.** Sample individual unit response curves (LRF models with $M = 256$).

**Figure supplement 3.** As in the main figure but for the RI models.

(4 Methods and Materials), only inward solutions exist for models with $M < 32$, and the outward solutions emerge more often as $M$ increases (*Figure 9—figure supplement 2*). Combined, these results with LRF and RI models indicate that the form and position of the nonlinearity in the circuit play a role in selecting between different optimized solutions. This suggests that further studies of the nonlinearities in LPLC2 processing will lead to additional insight into how a population of LPLC2s encodes looming stimuli.

The current binary prediction task is relatively easy for our loom detection models, as can be seen by the saturated AUC scores when $M$ is large (*Figure 9A*, *Figure 9—figure supplement 1A*). Thus, we engineered a new set of stimuli, where for the hit, miss, and retreat cases, we added a rotational background to increase the difficulty of the task. The object of interest, which is moving toward or away from the observer, also rotates with the background. This arrangement mimics self-rotation of the fly while observing a looming or retreating object in a cluttered background. To train an LRF model, we added such rotational distraction to half of the hit, miss, and retreat cases. The outward and inward solutions both persist (*Figure 9—figure supplement 3*), although in this case, outward solutions outperform inward ones (*Figure 9—figure supplement 3*). It remains unclear whether this added complexity brings our artificial stimuli closer to actual detection tasks performed by flies, but this result makes clear that identifying the natural statistics of loom will be important to understanding loom inferences.

## Activation patterns of computational solutions resemble biological responses

The outward solutions have a receptive field structure that is similar to LPLC2 neurons, based on anatomical and functional studies. However, it is not clear whether these models possess the functional properties of LPLC2 neurons, which have been studied systematically (*Klapoetke et al., 2017*; *Ache et al., 2019*). To see how trained units compare to LPLC2 neuron properties, we presented stimuli to the trained outward model solution to compare its responses to those measured in LPLC2 (*Figure 10*).

The outward unit behaves similarly to LPLC2 neurons on many different types of stimuli. Not surprisingly, the unit is selective for loom signals and does not have strong responses to non-looming signals (*Figure 10B*). Moreover, the unit closely follows the responses of LPLC2 neurons to various expanding bar stimuli, including the inhibitory effects of inward motion (*Figure 10C and D*). In addition, in experiments, motion signals that appear at the periphery of the receptive field suppress the activity of the LPLC2 neurons (periphery inhibition) (*Klapoetke et al., 2017*), and this phenomenon is successfully predicted by the outward unit (*Figure 10E and F*) due to its broad inhibitory filters (*Figure 10A*).

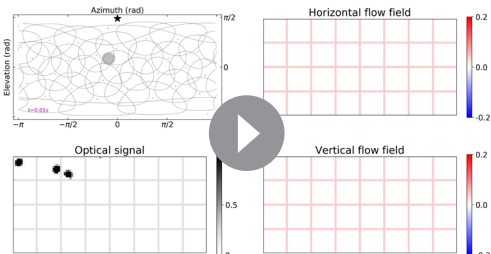

**Video 5.** Movie of unit responses for a hit stimulus (outward solution of the LRF model with 32 units). Top left panel: the same as in the top row of Figure 8A; bottom left, top right, bottom left panels: the same as in Video 1 but with more units. The movie has been slowed down by a factor of 10.

https://elifesciences.org/articles/72067/figures#video5

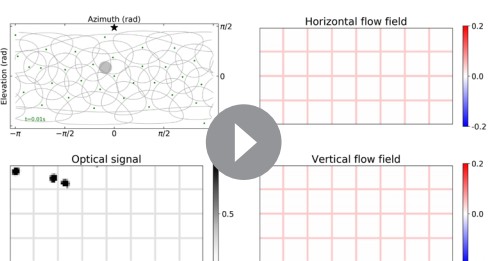

**Video 6.** Movie of unit responses for a hit stimulus (inward solution of the LRF model with 32 units). The same arrangement as Video 5 but for an inward model.

https://elifesciences.org/articles/72067/figures#video6

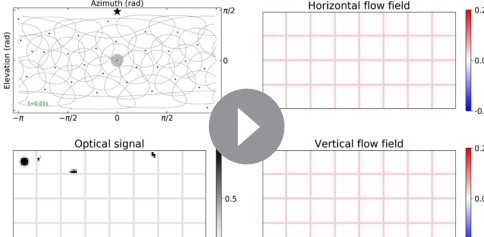

**Video 7.** Movie of unit responses for a miss stimulus (outward solution of the LRF model with 32 units). The same arrangement as Video 5.
https://elifesciences.org/articles/72067/figures#video7

**Video 8.** Movie of unit responses for a miss stimulus (inward solution of the LRF model with 32 units). The same arrangement as Video 6.
https://elifesciences.org/articles/72067/figures#video8

The unit also correctly predicts response patterns of the LPLC2 neurons for expanding bars with different orientations (**Figure 10G and H**).

The ratio of object size to approach velocity, or $R/v$, is an important parameter for looming stimuli, and many studies have investigated how the response patterns of loom-sensitive neurons depend on this ratio (Top panels in **Figure 10I, J, K and L**; **Gabbiani et al., 1999**; **von Reyn et al., 2017**; **Ache et al., 2019**; **de Vries and Clandinin, 2012**). Here, we presented the trained model (**Figure 10A**) with hit stimuli with different $R/v$ ratios, and compared its response with experimental measurements (**Figure 10I–L**). Surprisingly, although our model only has angular velocities as inputs (**Figure 3**), it reliably encodes the angular size of the stimulus rather than its angular velocity (**Figure 10J**). This is indicated by the collapsed response curves with different $R/v$ ratios (up to different scales) when plotted against the angular sizes (**von Reyn et al., 2017**). When the curves are plotted against angular velocity, they shift for different $R/v$ ratios, which means the response depends on the velocity $v$ of the object, since $R$ is fixed to be 1. The relative shifts in these curves are consistent with properties of LPLC2.

There are two ways that this angular size tuning likely arises. First, in hit stimuli, the angular size and angular velocity are strongly correlated (**Gabbiani et al., 1999**), which means the angular size affects the magnitude of the motion signals. Second, in hit stimuli, the angular size is proportional to the path length of the outward-moving edges. This angular circumference of the hit stimulus determines how many motion detectors are activated, so that integrated motion signal strength is related to the size. Both of these effects influence the response patterns of the model units (and the LPLC2 neurons). Beyond the tuning to stimulus size, the outward model also reproduces a canonical linear relationship between the peak response time relative to the collision and the $R/v$ ratio (**Figure 10L**; **Gabbiani et al., 1999**; **Ache et al., 2019**).

Not surprisingly, the inward solution cannot reproduce the neural data (**Figure 10—figure supplement 1**). On the other hand, some forms of the the RI model outward solutions can closely reproduce the neural data (**Figure 10—figure supplement 2**), while other outward solutions fail to do so (**Figure 10—figure supplement 3**, **Figure 10—figure supplement 4**). For example, some RI outward solutions predict the patterns in the wide expanding bars differently and out of phase from the

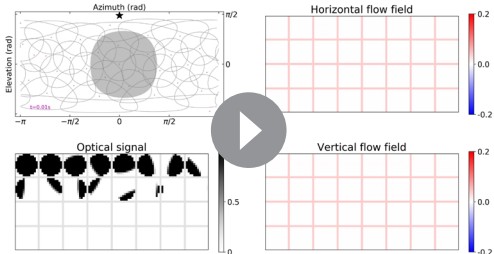

**Video 9.** Movie of unit responses for a retreat stimulus (outward solution of the LRF model with 32 units). The same arrangement as Video 5.
https://elifesciences.org/articles/72067/figures#video9

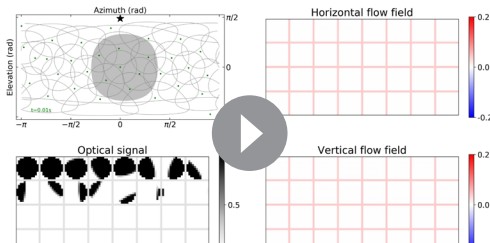

**Video 10.** Movie of unit responses for a retreat stimulus (inward solution of the LRF model with 32 units). The same arrangement as Video 6.
https://elifesciences.org/articles/72067/figures#video10

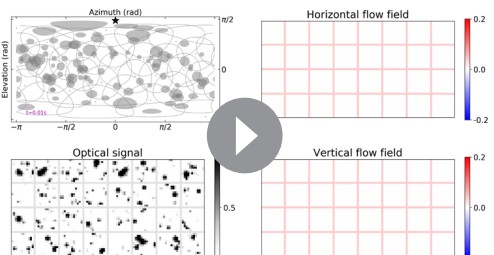

**Video 11.** Movie of unit responses for a rotation stimulus (outward solution of the LRF model with 32 units). The same arrangement as Video 5.
https://elifesciences.org/articles/72067/figures#video11

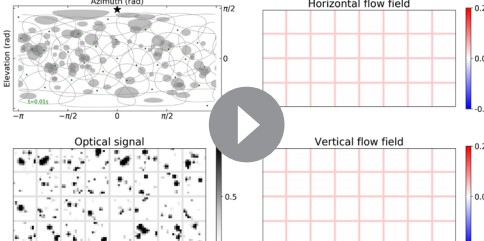

**Video 12.** Movie of unit responses for a rotation stimulus (inward solution of the LRF model with 32 units). The same arrangement as Video 6.
https://elifesciences.org/articles/72067/figures#video12

biological data (*Figure 10—figure supplement 3H*), do a poor job predicting the response curves of the LPLC2 neurons to looming signals with different $R/v$ ratios (*Figure 10—figure supplement 3J and K*), respond strongly to the moving gratings (*Figure 10—figure supplement 4B*), cannot show the peripheral inhibition (*Figure 10—figure supplement 4E and F*), and so on. This shows that, for the RI models, even within the family of learned outward solutions, there is variability in the learned response properties. Although solving the inference problem with the RI model obtains many of the neural response properties, additional constraints could be required. These additional constraints may be built into the LRF model, causing its outward solutions to more closely match neural responses.

## Discussion

In this study, we have shown that training a simple network to detect collisions gives rise to a computation that closely resembles neurons that are sensitive to looming signals. Specifically, we optimized a neural network model to detect whether an object is on a collision course based on visual motion signals (*Figure 3*), and found that one class of optimized solution matched the anatomy of motion

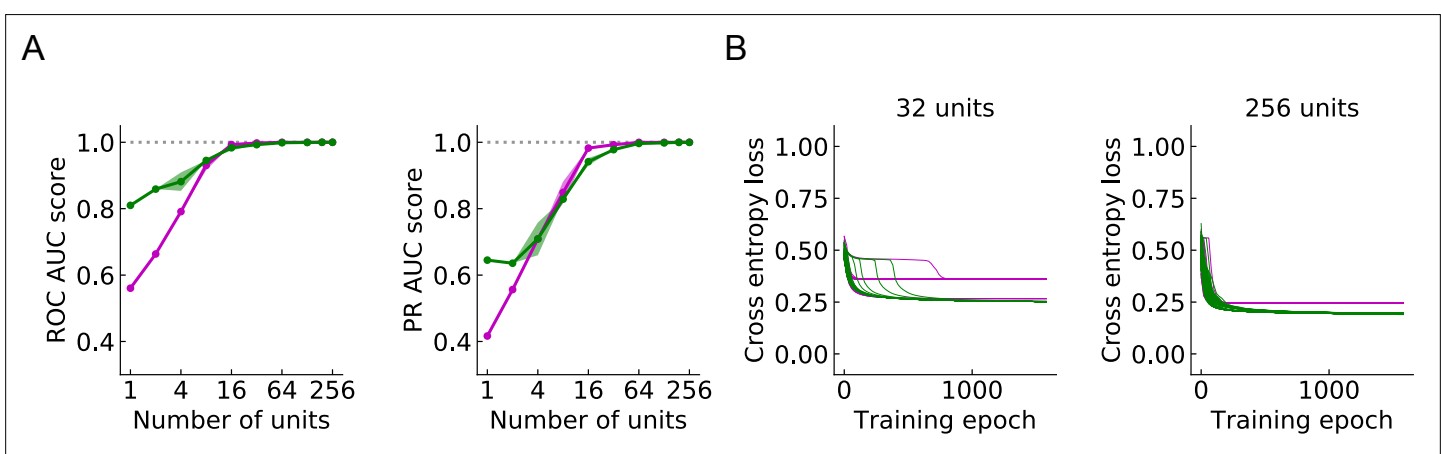

**Figure 9.** Large populations of units improve performance (LRF models) (Materials and methods). (**A**) Both ROC and PR AUC scores increase as the number of units increases. Colored lines and dots: average scores; shading: one standard deviation of the scores over the trained models. Magenta: outward solutions; green: inward solutions. The dotted horizontal gray lines indicate the value of 1. (**B**) As the population of units increases, cross entropy losses of the outward solutions approach the losses of the inward solutions.

The online version of this article includes the following figure supplement(s) for figure 9:

**Figure supplement 1.** As in the main figure but for RI models.

**Figure supplement 2.** The ratio of the number of the two types of solutions.

**Figure supplement 3.** As in the main figure but for LRF models trained using stimuli that include self-rotation during hits, misses, and retreats.

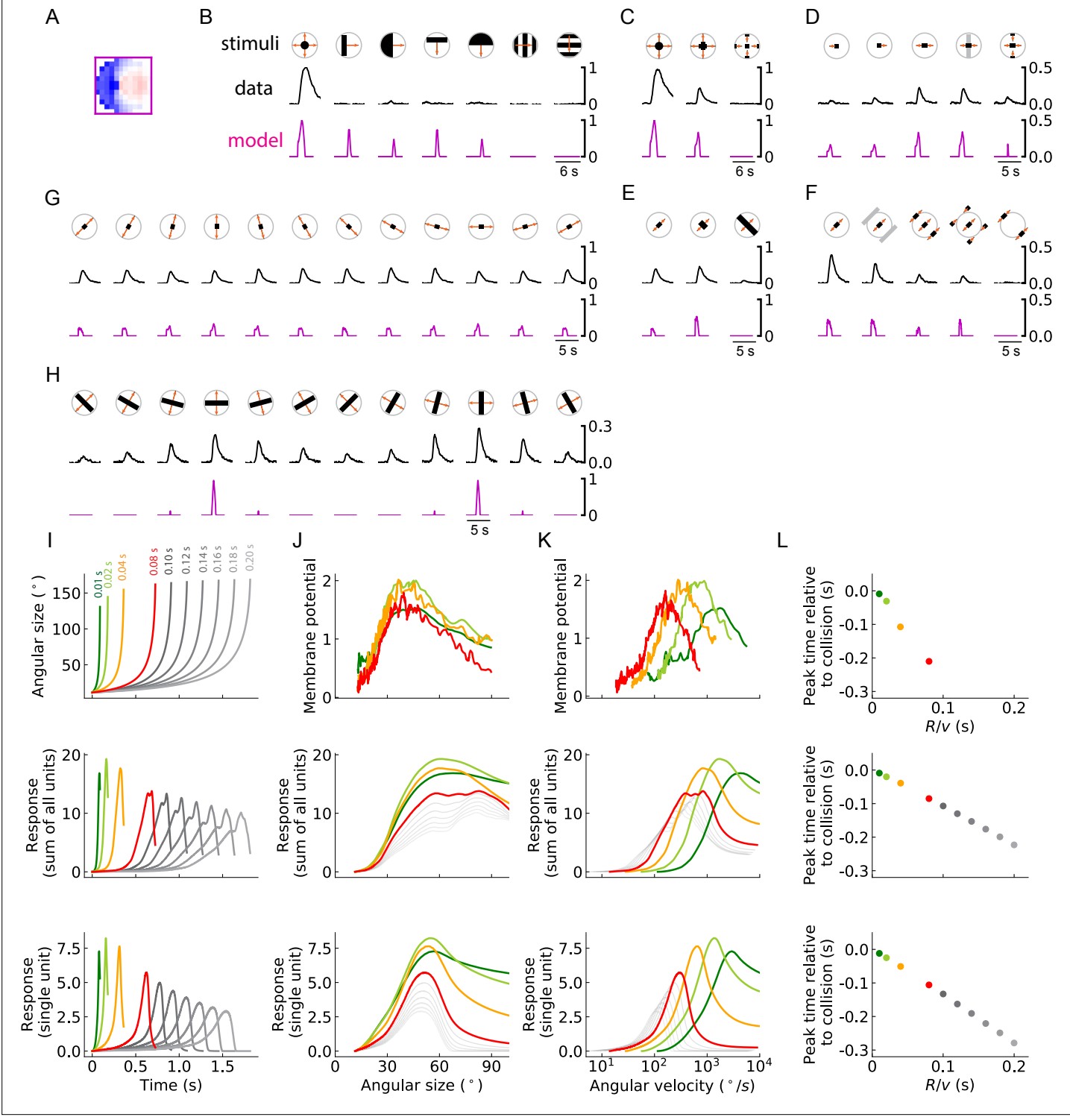

**Figure 10.** Units of models trained on binary classification tasks exhibit similar responses to LPLC2 neuron experimental measurements (outward solution of the LRF model with 256 units). (**A**) The trained filter. (**B–H**) Comparisons of the responses of the unit with the trained filter in (**A**) and LPLC2 neurons to a variety of stimuli (Materials and methods). Black lines: data (*Klapoetke et al., 2017*); magenta lines: LRF unit. Compared with the original plots (*Klapoetke et al., 2017*), all the stimulus icons here except the ones in (**B**) have been rotated 45 degrees to match the cardinal directions of LP layers as described in this study. All response curves are normalized by the peak value of the left most panel in (**B**). (**I**) Top: temporal trajectories of the angular sizes for different $R/v$ ratios (color labels apply throughout (**I–L**)) (Materials and methods). Middle: response as a function of time for the sum of all 256 units. Bottom: response as a function of time for one of the 256 units. (**J–L**) Top: experimental data (LPLC2/non-LC4 components of GF activity.

*Figure 10 continued on next page*

*Figure 10 continued*

Data from *von Reyn et al., 2017*; *Ache et al., 2019*). Middle: sum of all 256 units. Bottom: response of one of the 256 units. Responses as function of angular size (**J**), response as function of angular velocity (**K**), relationship between peak time relative to collision and *R/v* ratios (**L**). We considered the first peak when there were two peaks in the response, such as in the grey curves in the middle panel of (**I**).

The online version of this article includes the following figure supplement(s) for figure 10:

**Figure supplement 1.** As in the main figure but for an inward solution of the LRF model obtained from the same training procedure.

**Figure supplement 2.** As in the main figure but for an outward solution of the RI model with 256 units.

**Figure supplement 3.** As in the main figure but for a second outward solution of the RI model with 256 units.

**Figure supplement 4.** As in the main figure but for a third outward solution of the RI model with 256 units.

inputs to LPLC2 neurons (*Figures 1, 5 and 6*). Importantly, this solution reproduces a wide range of experimental observations of LPLC2 neuron responses (*Figure 10*; *Klapoetke et al., 2017*; *von Reyn et al., 2017*; *Ache et al., 2019*).

The radially structured dendrites of the LPLC2 neuron in the LP can account for its response to motion radiating outward from the receptive field center (*Klapoetke et al., 2017*). Our results show that the logic of this computation can be understood in terms of inferential loom detection by the *population* of units. In particular, for an individual detector unit, an inward structure can make a better loom detector than an outward structure, since it is sensitive to colliding objects originating from a wider array of incoming angles (*Figure 7*). As the number of units across visual space increases, the performance of the outward-sensitive receptive field structure comes to match the performance of the inward solutions (*Figure 9*, *Figure 9—figure supplement 1*, *Figure 9—figure supplement 2*). As the number of units increases, the inhibitory component of the outward solutions also becomes broader as the population size becomes larger, which is crucial for reproducing key experimental observations, such as peripheral inhibition (*Figure 10*; *Klapoetke et al., 2017*). The optimized solutions depend on the number of detectors, and this is likely related to the increasing overlap in receptive fields as the population grows (*Figure 6*). This result is consistent with prior work showing that populations of neurons often exhibit different and improved coding strategies compared to individual neurons (*Pasupathy and Connor, 2002*; *Georgopoulos et al., 1986*; *Vogels, 1990*; *Franke et al., 2016*; *Zylberberg et al., 2016*; *Cafaro et al., 2020*). Thus, understanding anatomical, physiological, and algorithmic properties of individual neurons can require considering the population response. The solutions we found to the loom inference problem suggest that individual LPLC2 responses should be interpreted in light of the population of LPLC2 responses.

Our results shed light on discussions of $\eta$-like (encoding angular size) and $\rho$-like (encoding angular velocity) looming sensitive neurons in the literature (*Gabbiani et al., 1999*; *Wu et al., 2005*; *Liu et al., 2011*; *Shang et al., 2015*; *Temizer et al., 2015*; *Dunn et al., 2016*; *von Reyn et al., 2017*; *Ache et al., 2019*). In particular, these optimized models clarify an interesting but puzzling fact: LPLC2 neurons transform their inputs of direction-selective motion signals to computations of angular size (*Ache et al., 2019*). Consistent with this tuning, our model also shows a linear relationship between the peak time relative to collision and the *R/v* ratio, which should be followed by loom sensitive neurons that encode angular size (*Peek and Card, 2016*). In both cases, these properties appear to be the simple result of training the constrained model to reliably detect looming stimuli.

The units of the outward solution exhibit sparsity in their responses to looming stimuli, in contrast to the denser representations in the inward solution (*Figure 8*). During a looming event, in an outward solution, most of the units are quiet and only a few adjacent units have very large activities, reminiscent of sparse codes that seem to be favored, for instance, in cortical encoding of visual scenes (*Olshausen and Field, 1996*; *Olshausen and Field, 1997*). Since the readout of our model is a summation of the activities of the units, sparsity does not directly affect the performance of the model, but is an attribute of the favored solution. For a model with a different loss function or with noise, the degree of sparsity might be crucial. For instance, the sparse code of the outward model might make it easier to localize a hit stimulus (*Morimoto et al., 2020*), or might make the population response more robust to noise (*Field, 1994*).

Experiments have shown that inhibitory circuits play an important role for the selectivity of LPLC2 neurons. For example, motion signals at the periphery of the receptive field of an LPLC2 neuron inhibit its activity. This peripheral inhibition causes various interesting response patterns of the LPLC2 neurons

to different types of stimuli (*Figure 10E and F*; *Klapoetke et al., 2017*). However, the structure of this inhibitory field is not fully understood, and our model provides a tool to investigate how the inhibitory inputs to LPLC2 neurons affect circuit performance on loom detection tasks. The strong inhibition on the periphery of the receptive field arises naturally in the outward solutions after optimization. The extent of the inhibitory components increases as more units are added to models (*Figure 6*). The broad inhibition appears in our model to suppress responses to the non-hit stimuli, and as in the data, the inhibition is broader than one might expect if the neuron were simply being inhibited by inward motion. These larger inhibitory fields are also consistent with the larger spatial pooling likely to be supplied by inhibitory LPi inputs (*Klapoetke et al., 2017*).

The synthetic stimuli used to train models in this study were unnatural in two ways. The first way was in the proportion of hits and non-hits. We trained with 25% of the training data representing hits. The true fraction of hits among all stimuli encountered by a fly is undoubtedly much less, and this affects how the loss function weights different types of errors. It is also clear that a false-positive hit (in which a fly might jump to escape an object not on collision course) is much less penalized during evolution than a false-negative (in which a fly doesn't jump and an object collides, presumably to the detriment of the fly). It remains unclear how to choose these weights in the training data or in the loss function, but they affect the receptive field weights optimized by the model.

The second issue with the stimuli is that they were caricatures of stimulus types, but did not incorporate the richness of natural stimuli. This richness could include natural textures and spatial statistics (*Ruderman and Bialek, 1994*), which seem to impact motion detection algorithms (*Fitzgerald and Clark, 2015*; *Leonhardt et al., 2016*; *Chen et al., 2019*). This richness could also include more natural trajectories for approaching objects. Another way to enrich the stimuli would be to add noise, either in inputs to the model or in the model's units themselves. We explored this briefly by adding self-rotation-generated background motion; under those conditions, both solutions were present but optimized outward solutions performed better than the inward solutions (*Figure 9—figure supplement 3*, Materials and methods). This indicates that the statistics of the stimuli may play an important role in selecting solutions for loom detection. However, it remains less clear what the true performance limits of loom detection are, since most experiments use substantially impoverished looming stimuli. Moreover, it is challenging to characterize the properties of natural looming events. An interesting future direction will be to investigate the effects of more complex and naturalistic stimuli on the model's filters and performance, as well as on LPLC2 neuron responses themselves.

For simplicity, our models did not impose the hexagonal geometry of the compound eye ommatidia. Instead, we assumed that the visual field is separated into a Cartesian lattice with $5°$ spacing, each representing a local motion detector with two spatially separated inputs (*Figure 3*). This simplification alters slightly the geometry of the motion signals compared to the real motion detector receptive fields (*Shinomiya et al., 2019*). This could potentially affect the learned spatial weightings and reproduction of the LPLC2 responses to various stimuli, since the specific shapes of the filters matter (*Figure 10*). Thus, the hexagonal ommatidial structure and the full extent of inputs to T4 and T5 might be crucial if one wants to make comparisons with the dynamics and detailed responses of LPLC2 neurons. However, this geometric distinction seems unlikely to affect the main results of how to infer the presence of hit stimuli.

Our model requires a field of estimates of the local motion. Here, we used the simplest model – the Hassenstein-Reichardt correlator model *Equation 3* (Materials and methods, *Hassenstein and Reichardt, 1956*) – but the model could be extended by replacing it with a more sophisticated model for motion estimation. Some biophysically realistic ones might take into account synaptic conductances (*Gruntman et al., 2018*; *Gruntman et al., 2019*; *Badwan et al., 2019*; *Zavatone-Veth et al., 2020*) and could respond to static features of visual scenes (*Agrochao et al., 2020*). Alternatively, in natural environments, contrasts fluctuate in time and space. Thus, if one includes more naturalistic spatial and temporal patterns, one might consider a motion detection model that could adapt to changing contrasts in time and space (*Drews et al., 2020*; *Matulis et al., 2020*).

Although the outward filter of the unit emerges naturally from our gradient descent training protocol, that does not mean that the structure is learned by LPLC2 neurons in the fly. There may be some experience dependent plasticity in the fly eye (*Kikuchi et al., 2012*), but these visual computations are likely to be primarily genetically determined. Thus, one may think of the computation of the LPLC2 neuron as being shaped through millions of years of evolutionary optimization. Optimization

algorithms at play in evolution may be able to avoid getting stuck in local optima (*Stanley et al., 2019*), and thus work well with the sort of shallow neural network found in the fly eye.

In this study, we focused on the motion signal inputs to LPLC2 neurons, and we neglected other inputs to LPLC2 neurons, such as those coming from the lobula that likely report non-motion visual features. It would be interesting to investigate how this additional non-motion information affects the performance and optimized solutions of the inference units. For instance, another lobula columnar neurons, LC4, is loom sensitive and receives inputs in the lobula (*von Reyn et al., 2017*). The LPLC2 and LC4 neurons are the primary excitatory inputs to the GF, which mediates escape behaviors (*von Reyn et al., 2014*; *Ache et al., 2019*). The inference framework set out here would allow one to incorporate parallel non-motion intensity channels, either by adding them into the inputs to the LPLC2-like units, or by adding in a parallel population of LC4-like units. This would require a reformulation of the probabilistic model in *Equation 6*. Notably, one of the most studied loom detecting neurons, the lobula giant movement detector (LGMD) in locusts, does not appear to receive direction-selective inputs, as LPLC2 does (*Rind and Bramwell, 1996*; *Gabbiani et al., 1999*). Thus, the inference framework set out here could be flexibly modified to investigate loom detection under a wide variety of constraints and inputs, which allow it to be applied to other neurons, beyond LPLC2.

## Materials and methods
### Code availability
Code to perform all simulations in this paper and to reproduce all figures is available at https://github.com/ClarkLabCode/LoomDetectionANN, (copy archived at swh:1:rev:864fd3d591bc9e3923189320d7197bdd0cd85448; *Zhou, 2021*).

### Coordinate system and stimuli
We designed a suite of visual stimuli to simulate looming objects, retreating objects, and rotational visual fields. In this section, we describe the suite of stimuli and the coordinate systems used in our simulations (*Figure 4—figure supplement 1*).

In our simulations and training, the fly is at rest on a horizontal plane, with its head pointing in a specific direction. The fly head is modeled to be a point particle with no volume. A three-dimensional right-handed frame of reference $\Sigma$ is set up and attached to the fly head at the origin. The $z$ axis points in the anterior direction from the fly head, perpendicular to the line that connects the two eyes, and in the horizontal plane of the fly; the $y$ axis points toward the right eye, also in the horizontal plane; and the $x$ axis points upward and perpendicular to the horizontal plane. Looming or retreating objects are represented in this space by a sphere with radius $R = 1$, and the coordinates of an object's center at time $t$ are denoted as $\mathbf{r}(t) = (x(t), y(t), z(t))$. Thus, the distance between the object center and the fly head is $D(t) = \|\mathbf{r}(t)\| = \sqrt{x^2(t) + y^2(t) + z^2(t)}$.

Within this coordinate system, we set up cones to represent individual units. The receptive field of LPLC2 neurons is measured at roughly $60°$ in diameter (*Klapoetke et al., 2017*). Thus, we here model each unit as a cone with its vertex at the origin and with half-angle of $30°$. For each unit $m$ ($m = 1, 2, \ldots, M$), we set up a local frame of reference $\Sigma_m$ (*Figure 4—figure supplement 1*): the $z_m$ axis is the axis of the cone and its positive direction points outward from the origin. The local $\Sigma_m$ can be obtained from $\Sigma$ by two rotations: around $x$ of $\Sigma$ and around the new $y'$ after the rotation around $x$. For each unit, its cardinal directions are defined as: upward (positive direction of $x_m$), downward (negative direction of $x_m$), leftward (negative direction of $y_m$). and rightward (positive direction of $y_m$). To get the signals that are received by a specific unit $m$, the coordinates of the object in $\Sigma$ are rotated to the local frame of reference $\Sigma_m$.

Within this coordinate system, we can set up cones representing the extent of a spherical object moving in the space. The visible outline of a spherical object spans a cone with its point at the origin. The half-angle of this cone is a function of time and can be denoted as $\theta_s(t)$:

$$\theta_s(t) = \arcsin \frac{R}{D(t)}. \tag{1}$$

One can calculate how the cone of the object overlaps with the receptive field cones of each unit.

There are multiple layers of processing in the fly visual system (*Takemura et al., 2017*), but here we focus on two coarse grained stages of processing: (1) the estimation of local motion direction from optical intensities by motion detection neurons T4 and T5 and (2) the integration of the flow fields by LPLC2 neurons. In our simulations, the interior of the $m$th unit cone is represented by a $N$-by-$N$ matrix, so that each element in this matrix (except the ones at the four corners) indicates a specific direction in the angular space within the unit cone. If an element also falls within the object cone, then its value is set to 1; otherwise it is 0. Thus, at each time $t$, this matrix is an optical intensity signal and can be represented by $C(x_m, y_m, t)$, where $(x_m, y_m)$ are the coordinates in $\Sigma_m$. In general, $N$ should be large enough to provide good angular resolutions. Then, $K^2$ ($K$) motion detectors are evenly distributed within the unit cone, with each occupying an $L$-by-$L$ grid in the $N$-by-$N$ matrix, where $L = N/K$. This $L$-by-$L$ grid represents a $5°$-by-$5°$ square in the angular space, consistent with the approximate spacing of the inputs of motion detectors T4 and T5. This arrangement effectively uses high spatial resolution intensity data to compute local intensity before it is discretized into motion signals with a resolution of $5°$. Since the receptive field of an LPLC2 neuron is roughly $60°$, the value of $K$ is chosen to be 12. To get sufficient angular resolution for the local motion detectors, $L$ is set to be 4, so that $N$ is set to 48.

Each motion detector is assumed to be a Hassenstein Reichardt Correlator (HRC) and calculates local flow fields from $C(x_m, y_m, t)$ (*Hassenstein and Reichardt, 1956*; *Figure 3—figure supplement 1*). The HRC used here has two inputs, separated by $5°$ in angular space. Each input applies first a spatial filter on the contrast $C(x_m, y_m, t)$ and then temporal filters:

$$I_j(t; x_m, y_m) = \sum_{t'=0}^{t} \sum_{x'_m=-N}^{N} \sum_{y'_m=-N}^{N} f_j(t')G(x'_m, y'_m)C(x_m - x'_m, y_m - y'_m, t - t'), \tag{2}$$

where $f_j$ ($j \in 1, 2$) is a temporal filter and $G$ is a discrete 2d Gaussian kernel with mean $0°$ and standard deviation of $2.5°$ to approximate the acceptance angle of the fly photoreceptors (*Stavenga, 2003*). The temporal filter $f_1$ was chosen to be an exponential function $f_1(t') = (1/\tau)\exp(-t'/\tau)$ with $\tau$ set to 0.03 seconds (*Salazar-Gatzimas et al., 2016*), and $f_2$ a delta function $f_2 = \delta(t')$. This leads to

$$F(t; x_{m1}, y_{m1}, x_{m2}, y_{m2}) = I_1(t; x_{m1}, y_{m1})I_2(t; x_{m2}, y_{m2}) - I_1(t; x_{m2}, y_{m2})I_2(t; x_{m1}, y_{m1}). \tag{3}$$

as the local flow field at time $t$ between two inputs located at $(x_{m1}, y_{m1})$ and $(x_{m2}, y_{m2})$.

Four types of T4 and T5 neurons have been found that project to layers 1, 2, 3, and 4 of the LP. Each type is sensitive to one of the cardinal directions: down, up, left, right (*Maisak et al., 2013*). Thus, in our model, there are four non-negative, local flow fields that serve as the only inputs to the model: $U_-(t)$ (downward, corresponding LP layer 4), $U_+(t)$ (upward, LP layer 3), $V_-(t)$ (leftward, LP layer 1), and $V_+(t)$ (rightward, LP layer 2), each of which is a $K$-by-$K$ matrix. To calculate these matrices, two sets of motion detectors are needed, one for the vertical directions and one for the horizontal directions. The HRC model in *Equation 3* is direction sensitive and is opponent, meaning that for motion in the preferred (null) direction, the output of the HRC model is positive (negative). Thus, assuming that upward (rightward) is the preferred vertical (horizontal) direction, we obtain the non-negative elements of the four flow fields as

$$\begin{aligned}
[U_-(t)]_{k_1 k_2} &= \max(0, -F(t; x_{m1}, y_m, x_{m2}, y_m)) \\
[U_+(t)]_{k_1 k_2} &= \max(0, F(t; x_{m1}, y_m, x_{m2}, y_m)) \\
[V_-(t)]_{k_1 k_2} &= \max(0, -F(t; x_m, y_{m1}, x_m, y_{m2})) \\
[V_+(t)]_{k_1 k_2} &= \max(0, F(t; x_m, y_{m1}, x_m, y_{m2})),
\end{aligned}$$

where $k_1, k_2 \in \{1, 2, \ldots, K\}$. In the above expressions, for $[U_-(t)]_{k_1 k_2}$ and $[U_+(t)]_{k_1 k_2}$, the vertical motion detector at $(k_1, k_2)$ has its two inputs located at $(x_{m1}, y_m)$ and $(x_{m2}, y_m)$, respectively. Similarly, for for $[V_-(t)]_{k_1 k_2}$ and $[V_+(t)]_{k_1 k_2}$, the horizontal motion detector at $(k_1, k_2)$ has its two inputs located at $(x_m, y_{m1})$ and $(x_m, y_{m2})$. Using the opponent HRC output as the motion signals for each layer is reasonable because the motion detectors T4 and T5 are highly direction-selective over a large range of inputs (*Maisak et al., 2013*; *Creamer et al., 2018*) and synaptic, 3-input models for T4 are approximately equivalent to opponent HRC models (*Zavatone-Veth et al., 2020*).

We simulated the trajectories $\mathbf{r}(t)$ of the object in the frame of reference $\Sigma$ at a time resolution of 0.01 s, which is also be the time step of the training and testing stimuli. For hit, miss, and retreat

cases, the trajectories of the object are always straight lines, and the velocities of the object were randomly sampled from a range $[2R, 10R](s^{-1})$ with the trajectories confined to be within a sphere of $5R$ centered at the fly head. The radius of the object, $R$, is always set to be one except in the rotational stimuli. To generate rotational stimuli, we placed 100 objects with various radii selected uniformly from $[0, 1]$ at random distances ($[5, 15]$) and positions around the fly, and rotated them all around a randomly chosen axis. The rotational speed was chosen from a Gaussian distribution with mean $0°/s$ and standard deviation $200°/s$, a reasonable rotational velocity for walking flies (*DeAngelis et al., 2019*). In one case, training data included both an object moving and global rotation due to self-rotation (*Figure 9—figure supplement 3*). These stimuli were simply combinations of the rotational stimuli with the other three cases (hit, miss, and retreat), so that the object that moves in the depth dimension also rotates together with the background.

We reproduced a range of stimuli used in a previous study (*Klapoetke et al., 2017*) and tested them on our trained model (*Figure 10B–H*). To match the cardinal directions of LP layers (*Figure 1*), we have rotated the stimuli (except in *Figure 10B*) 45 degrees compared with the ones displayed in the figures in *Klapoetke et al., 2017*. The disc (*Figure 10B and C*) expands from $20°$ to $60°$ with an edge speed of $10°/s$. All the bar and edge motions have an edge speed of $20°/s$. The width of the bars are $60°$ (right panel of *Figure 10E and H*), $20°$ (middle panel of *Figure 10E*), and $10°$ (all the rest). All the responses of the models (except in *Figure 10B*) have been normalized by the peak of the response to the expanding disc (*Figure 10B*).

We created a range of hit stimuli with various $R/v$ ratios: $0.01\,s$, $0.02\,s$, $0.04\,s$, $0.08\,s$, $0.10\,s$, $0.12\,s$, $0.14\,s$, $0.16\,s$, $0.18\,s$, $0.20\,s$. The radius $R$ of the spherical object is fixed to be 1, and the velocity is changed accordingly to achieve different $R/v$ ratios.

## Models

LPLC2 neurons have four dendritic structures in the four LP layers, and they receive direct excitatory inputs from T4/T5 motion detection neurons (*Maisak et al., 2013*; *Klapoetke et al., 2017*). It has been proposed that each dendritic structure also receives inhibitory inputs mediated by lobula plate intrinsic interneurons, such as LPi4-3 (*Klapoetke et al., 2017*). We built two model units to approximate this anatomy.

LRF models A linear receptive field (LRF) model is characterized by a real-valued filter, represented by a 12-by-12 matrix $W^{\mathfrak{r}}$ (*Figure 4A*). The elements of the filter combine the effects of the excitatory and inhibitory inputs and can take both positive (stronger excitation) and negative (stronger inhibition) values. We rotate $W^{\mathfrak{r}}$ counterclockwise by multiples of $90°$ to obtain the filters that are used to integrate the four motion signals: $U_-(t)$, $U_+(t)$, $V_-(t)$, $V_+(t)$. Specifically, we define the corresponding four filters as: $W^{\mathfrak{r}}_{U_-} = \text{rotate}(W^{\mathfrak{r}}, 270°)$, $W^{\mathfrak{r}}_{U_+} = \text{rotate}(W^{\mathfrak{r}}, 90°)$, $W^{\mathfrak{r}}_{V_-} = \text{rotate}(W^{\mathfrak{r}}, 180°)$, $W^{\mathfrak{r}}_{V_+} = \text{rotate}(W^{\mathfrak{r}}, 0°)$. In addition, we impose mirror symmetry on the filters, and with the above definitions of the rotated filters, the upper half of $W^{\mathfrak{r}}$ is a mirror image of the lower half of $W^{\mathfrak{r}}$. Thus, there are in total 72 parameters in the filters. In fact, since only the elements within a 60 degree cone contribute to the filter for the units, the corners are excluded, resulting in only 56 trainable parameters.

In computer simulations, the filters or weights and flow fields are flattened to be one-dimensional column vectors. The response of a single LRF unit $m$ is:

$$r_m(t) = \phi\left( (W^{\mathfrak{r}}_{U_-})^T U_-(t) + (W^{\mathfrak{r}}_{U_+})^T U_+(t) + (W^{\mathfrak{r}}_{V_+})^T V_+(t) + (W^{\mathfrak{r}}_{V_-})^T V_-(t) + b_r \right), \qquad (4)$$

where $\phi(\cdot) = \max(\cdot, 0)$ is the rectified linear unit (ReLU), and $b^{\mathfrak{r}} \in \mathbb{R}$ is the intercept (*Figure 4A*). The ReLU is used as the activation function in all the figures, except in one panel (*Figure 9—figure supplement 2*), where an exponential linear unit is used:

$$\phi_{\text{ELU}}(x) = \begin{cases} x, & x > 0 \\ e^x - 1, & x \leq 0. \end{cases}$$

For $M = 1$, the LRF model is very close to a generalized linear model, except that it includes an additional activation function $\phi$. This activation function changes the convexity of the model to make it a non-convex optimization problem, in general.

RI models The rectified inhibition (RI) models have two types of nonnegative filters, one excitatory and one inhibitory, represented by $W^e$ and $W^i$, respectively (*Figure 4B*). Each filter is a 12-by-12 matrix. The same rotational and mirror symmetries are imposed as in the LRF models, which leads to four excitatory filters as: $W^e_{U_-} = \text{rotate}(W^e, 270°)$, $W^e_{U_+} = \text{rotate}(W^e, 90°)$, $W^e_{V_-} = \text{rotate}(W^e, 180°)$, $W^e_{V_+} = \text{rotate}(W^e, 0°)$, and four inhibitory filters as: $W^i_{U_-} = \text{rotate}(W^i, 270°)$, $W^i_{U_+} = \text{rotate}(W^i, 90°)$, $W^i_{V_-} = \text{rotate}(W^i, 180°)$, $W^i_{V_+} = \text{rotate}(W^i, 0°)$. Thus, there are in total 112 parameters in the two sets of filters, excluding the elements in the corners.

The responses of the inhibitory units are:

$$
\begin{aligned}
r^i_{U_-}(t) &= \phi\left((W^i_{U_-})^T U_-(t) + b^i\right) \\
r^i_{U_+}(t) &= \phi\left((W^i_{U_+})^T U_+(t) + b^i\right) \\
r^i_{V_+}(t) &= \phi\left((W^i_{V_+})^T V_+(t) + b^i\right) \\
r^i_{V_-}(t) &= \phi\left((W^i_{V_-})^T V_-(t) + b^i\right),
\end{aligned}
$$

where $\phi(\cdot) = \max(\cdot, 0)$ is the ReLU, and $b^i \in \mathbb{R}$ is the intercept. In the RI model, the rectification of each inhibitory layer is motivated by the LPi neurons, which mediate the inhibition within each layer, and could themselves rectify their inhibitory input into LPLC2. The response of a single RI unit $m$ is

$$
r_m(t) = \phi\Bigg((W^e_{U_-})^T U_-(t) + (W^e_{U_+})^T U_+(t) + (W^e_{V_+})^T V_+(t) + (W^e_{V_-})^T V_-(t) - \\
\left(r^i_{U_-}(t) + r^i_{U_+}(t) + r^i_{V_+}(t) + r^i_{V_-}(t)\right) + b^e\Bigg),
$$

$$(5)$$

where $b^e \in \mathbb{R}$ is the intercept (*Figure 4B*). Interestingly, the RI models become equivalent to the LRF models if we remove the ReLU in the inhibitions and define $W^r_{U_-} = W^e_{U_-} - W^i_{U_-}$, $W^r_{U_+} = W^e_{U_+} - W^i_{U_+}$, $W^r_{V_+} = W^e_{V_+} - W^i_{V_+}$, $W^r_{V_-} = W^e_{V_-} - W^i_{V_-}$ and $b^r = b^e - 4b^i$.

For both the LRF and RI models, the inferred probability of hit for a specific trajectory is

$$
\widehat{P}_{\text{hit}} = \frac{1}{T} \sum_{t=1}^{T} \sigma\left(\sum_m r_m(t) + b\right),
$$

$$(6)$$

where $T$ is the total number of time steps in the trajectory and $\sigma(\cdot)$ is the sigmoid function. Since we are adding two intercepts ($b^r$, and $b$) to the LRF models, and three intercepts ($b^i$, $b^e$, and $b$) to the RI models, there are 58 and 115 parameters to train the two models, respectively.

## Training and testing

We created a synthetic data set containing four types of motion: *loom-and-hit*, *loom-and-miss*, *retreat*, and *rotation*. The proportions of these types were 0.25, 0.125, 0.125, and 0.5, respectively. In total, there were 5200 trajectories, with 4000 for training and 1200 for testing. Trajectories with motion type *loom-and-hit* are labeled as hit or $y_n = 1$ (probability of hit is 1), while trajectories of other motion types are labeled as non-hit or $y_n = 0$ (probability of hit is 0), where $n$ is the index of each specific sample. Models with smaller $M$ have fewer trajectories in the receptive field of any unit. For stability of training, we therefore increased the number of trajectories by factors of eight, four, and two for $M = 1, 2, 4$, respectively.

The loss function to be minimized in our training was the cross entropy between the label $y_n$ and the inferred probability of hit $\widehat{P}_{\text{hit}}$, and averaged across all samples, together with a regularization term:

$$
\text{cross entropy loss} = -\frac{1}{N} \sum_{n=1}^{N} \left\{ y_n \log \widehat{P}_{\text{hit}}(n) + (1 - y_n) \log(1 - \widehat{P}_{\text{hit}}(n)) \right\} + \beta \sum_W \|W\|^2,
$$

$$(7)$$

where $\widehat{P}_{\text{hit}}(n)$ is the inferred probability of hit for sample $n$, $\beta$ is the strength of the $\ell_2$ regularization, and $W$ represents all the effective parameters in the two excitatory and inhibitory filters.

The strength of the regularization $\beta$ was set to be $10^{-4}$, which was obtained by gradually increasing $\beta$ until the performance of the model on test data started to drop. The regularization sped up convergence of solutions, but the regularization strength did not strongly influence the main results in the paper.

To speed up training, rather than taking a temporal average as shown in *Equation 6*, a snapshot was sampled randomly from each trajectory, and the probability of hit of this snapshot was used to represent the whole trajectory, that is, $\widehat{P}_{\text{hit}} = \sigma \left( \sum_m r_m(t) + b \right)$, where $t$ is a random sample from $\{1, 2, \dots, T\}$. Mini-batch gradient descent was used in training, and the learning rate was 0.001.

After training, the models were tested on the entire trajectories with the probability of hit defined in *Equation 6*. Models trained only on snapshots performed well on the test data. During testing, the performance of the model was evaluated by the area under the curve (AUC) of the receiver operating characteristic (ROC) and precision-recall (PR) curves (*Hanley and McNeil, 1982*; *Davis and Goadrich, 2006*). TensorFlow (*Abadi et al., 2016*) was used to train all models.

## Clustering the solutions

We used the following procedure to cluster the solutions. For the LRF models, the filter of each solution was simply flattened to form a vector. But, for the RI mdoels, each solution had an excitatory and an inhibitory filter. We flattened these two filters, and concatenated them into a single vector. (The elements at the corners were deleted since they are outside of the receptive field.) Thus, each solution was represented by a vector, from which we calculated the cosine distance for each pair of solutions. The obtained distance matrix was then fed into a hierarchical clustering algorithm (*Virtanen et al., 2020*). After obtaining the hierarchical clustering, the outward and inward filters were identified by their shape. We counted the positive filter elements corresponding to flow fields with components radiating outward and subtracted the number of positive filter elements corresponding to flow fields with components directed inward. If the resulting value was positive, the filters were labeled as outward; otherwise, the filters were labeled as inward. We could also sum the elements rather than count them, but we found that the latter was more robust. If the elements in the concatenated vector were all close to zero, then the corresponding filters were labeled as zero solutions.

## Statistics

To calculate the fraction of active units for the model with $M = 256$ (*Figure 8B*), we examined the response curves of each unit to all trajectories of a specific type of stimuli. If a unit response is above baseline (dotted lines in *Figure 7B*), then the unit is counted as active. For each trajectory/stimulus, we obtained the number of active units. We used this number to calculate the mean and standard deviation of active units across all the trajectories within each type of stimulus (hit, miss, retreat, rotation).

For a model with $M$ units, where $M \in \{1, 2, 4, 8, 16, 32, 64, 128, 192, 256\}$, 200 random initializations were used to train it. For the LRF models, within these 200 training runs, the number of outward solutions $N_{\text{out}}$ were (starting from smaller values of $M$) 45, 59, 60, 65, 52, 63, 52, 57, 58, 49, and the number of inward solutions $N_{\text{in}}$ were 142, 135, 127, 119, 139, 133, 130, 118, 123, 119. For the RI models, the number of outward solutions $N_{\text{out}}$ were 44, 46, 48, 50, 48, 50, 53, 55, 58, 64, and the number of inward solutions $N_{\text{in}}$ were 39, 40, 39, 46, 53, 51, 35, 38, 12, 10. The average score curves and points in *Figure 9A*, *Figure 9—figure supplement 1A* and *Figure 9—figure supplement 2A* were obtained by taking the average among each type of solution, with the shading indicating the standard deviations. The curves and point in *Figure 9—figure supplement 1C* are the ratio of the number of outward solutions to the number of inward solutions. To obtain error bars (grey shading), we considered the training results as a binomial distribution, with the probability of obtaining an outward solution being $N_{\text{out}}/(N_{\text{out}} + N_{\text{in}})$, and with the probability of obtaining an inward solution being $N_{\text{in}}/(N_{\text{out}} + N_{\text{in}})$. Thus, the standard deviation of this binomial distribution is $\sigma_{\text{b}} = \sqrt{N_{\text{out}} N_{\text{in}}/(N_{\text{out}} + N_{\text{in}})}$. From this, we calculate the error bars as the propagated standard deviation (*Morgan et al., 1990*):

$$\text{propagated error} = \frac{N_{\text{out}}}{N_{\text{in}}} \sqrt{\left( \frac{\sigma_{\text{b}}}{N_{\text{out}}} \right)^2 + \left( \frac{\sigma_{\text{b}}}{N_{\text{in}}} \right)^2}. \tag{8}$$

## Acknowledgements

Research supported in part by NSF grants DMS-1513594, CCF-1839308, DMS-2015397, NIH R01EY026555, a JP Morgan Faculty Research Award, and the Kavli Foundation. We thank G Card and N Klapoetke for sharing data traces from their paper. We thank members of the Clark lab for discussions and comments.

## Additional information

### Funding

| Funder | Grant reference number | Author |
| --- | --- | --- |
| National Institutes of Health | R01EY026555 | Baohua Zhou<br>Damon A Clark |
| National Science Foundation | CCF-1839308 | Baohua Zhou<br>John Lafferty<br>Damon A Clark |
| National Science Foundation | DMS-1513594 | John Lafferty |
| Kavli Foundation | | John Lafferty |

The funders had no role in study design, data collection and interpretation, or the decision to submit the work for publication.

### Author contributions

Baohua Zhou, Conceptualization, Formal analysis, Investigation, Methodology, Software, Validation, Visualization, Writing – original draft, Writing – review and editing; Zifan Li, Conceptualization, Formal analysis, Investigation, Methodology, Software, Visualization, Writing – original draft; Sunnie Kim, Conceptualization, Formal analysis, Software; John Lafferty, Conceptualization, Funding acquisition, Methodology, Supervision, Writing – original draft, Writing – review and editing; Damon A Clark, Conceptualization, Funding acquisition, Methodology, Supervision, Visualization, Writing – original draft, Writing – review and editing

### Author ORCIDs

Baohua Zhou (ID) http://orcid.org/0000-0002-2627-9447
Sunnie Kim (ID) http://orcid.org/0000-0002-8901-7233
Damon A Clark (ID) http://orcid.org/0000-0001-8487-700X

### Decision letter and Author response

Decision letter https://doi.org/10.7554/eLife.72067.sa1
Author response https://doi.org/10.7554/eLife.72067.sa2

## Additional files

### Supplementary files

• Transparent reporting form

### Data availability

Code to perform all simulations in this paper and to reproduce all figures is available at https://github.com/ClarkLabCode/LoomDetectionANN, (copy archived at swh:1:rev:864fd3d591bc9e3923189320d7197bdd0cd85448).

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
