## [Editor Report]

This paper trains a simple neural network model to perform a behaviorally important task: the detection of looming objects. Two solutions emerge, one of which shares several properties with the actual circuit. This is a nice demonstration that training a CNN on a behaviorally-relevant task can reveal how the underlying computations work.

---

## [Decision Letter]

**Decision letter after peer review:**

Thank you for submitting your article "Shallow neural networks trained to detect collisions recover features of visual loom-selective neurons" for consideration by *eLife*. Your article has been reviewed by 3 peer reviewers, including Fred Rieke as Reviewing Editor and Reviewer #1, and the evaluation has been overseen by Ronald Calabrese as the Senior Editor. The following individual involved in review of your submission has agreed to reveal their identity: Catherine von Reyn (Reviewer #3).

Essential revisions:

The followed issues emerged in review – and were agreed upon by all of the reviewers in consultations.

1. Questions about the model architecture. Several model components (rotation and symmetry) were imposed rather than learned. Was this necessary? Can the model make (testable) predictions about connectomics data?

2. Types of solutions. The text and results needs to explore all three types of solution (inward, outward and unstructured) in more detail. It is currently difficult to understand why the inward and unstructured solutions are essentially dropped part way through.

3. More challenging tests of the model. Can you add distracting optic flow to the current stimulus set and/or use more naturalistic stimuli? This could help reduce the number of viable solutions.

4. Inhibitory component of the model. Inhibition is assumed to have specific properties (e.g. rectification) – and it is not clear if these are essential. Further, it is absent in some solutions. Are the properties of inhibition (when present) consistent with the broad LPi receptive fields?

5. Comparison of model with neural data. A stronger rationale is needed for why two of the many outward models are selected for comparison with neural data (and why comparisons are not made for the inward or unstructured models). It is also important to quantify the similarity of the models with neural data.

*Reviewer #1 (Recommendations for the authors):*

Line 26-27: It would be helpful to make a somewhat more general statement about the power of the approach that you take here.

Figure 3 is the first figure referred to, so moving it up to Figure 1 would make reading easier.

Line 79: clarify here you mean object motion, not motion of one of the edges.

Line 94-95: the relationship between timing and size-to-speed ratio is likely hard for most readers to make sense of here – suggest deleting.

Lines 150-151: suggest clarifying that excitation and inhibition in the model are not constrained to have opposite spatial dependencies as depicted in the Figure 4.

Line 170: suggest describing the loss function in a sentence in the Results.

Lines 174-176: It would be helpful to connect the outward and inward model terminology more clearly to the flow fields in Figure 3 here. I think this is just a matter of highlighting which elements of the grid in Figure 3 are relevant for each model.

Lines 177-178: describe performance measures here qualitatively.

Lines 206-209: the reason for the difference in baseline activity is not clear – and it requires a lot of effort to extract that from the methods. Can you give more intuition here in the results?

Lines 336-340: this is helpful, and some of it could come up earlier in the Results. More generally, it would be helpful to be clearer (especially in results) how much of the encoding of angular size is a property of expansion of the stimulus, and how much of how the computation is implemented.

*Reviewer #2 (Recommendations for the authors):*

– The manuscript is a bit difficult to understand. The authors may want to improve their explanations and figures to make them more accessible. For example, in Figure 7B, I can barely see the responses and don't see any grey lines. Perhaps showing only a subset of responses would make the figure clearer -- less is more.

– The usage of the term "ballistic" in the introduction is confusing. In many contexts, "ballistic" suggests free-falling motion; in this paper, the authors are referring to the distinction between ballistic and diffusive motion. To avoid confusion, I would suggest not using the term ballistic at all; instead, "straight line" or "linear" is just as expressive.

– The first figure that is cited in the text is Figure 3. I suggest reorganizing either the text or the figures so that the first figure that is cited is Figure 1.

– Figure 5, panel D: why are there two magenta curves?

– I would also suggest a careful reading to screen for typos -- I found a dozen or so, from misspelled words to mismatched parentheses.

*Reviewer #3 (Recommendations for the authors):*

1. Suggestions for improved or additional experiments, data or analyses:

a. The authors should provide their criteria for selecting a particular solution to compare to neural data.

b. The authors should evaluate how well their solutions predict neural data.

c. The authors need to mention that certain outward solutions have no inhibitory component (see Figure 5C, Figure 6 supplement 2). It needs to be discussed in the text and it would be very interesting to see how well these solutions recreate actual data.

d. It would be helpful for the authors to provide an example of an "unstructured" solution and an evaluation of its performance, even if it is included as a supplemental figure.

2. Recommendations for improving writing and presentation

a. Lines 89-90 – this can be better supported by adding the criteria/evaluation mentioned above.

b. Methods (~ line 483) – How is the HRC model using T5 (off) and T4 (on) motion input?

c. Lines 492-502 – What was the frame rate (timestep) for both training and testing stimuli?

d. Figures – Please increase the size when there is white space available. Make sure the pink and green color scheme for the two solution sets are very obvious.

e. Figure 1 caption – approximately half of the 200 LPLC2 are directly synaptic to the GF.

f. Figure 5 – is cross entropy loss the same as what is referred to as the loss function (equation 6) in the methods? If so, keep consistent. If not, please explain.

g. Figure 8D, it is difficult to see the boxplots.

h. Figure 10 I-L, it is difficult at first glance to realize what is neural data vs model output. Maybe label the rows instead?

i. Supplemental Figure 1. Add a schematic for the HRC model for readers who may not be familiar with it.

---

## [Author Response]

Essential revisions:The followed issues emerged in review – and were agreed upon by all of the reviewers in consultations.1. Questions about the model architecture. Several model components (rotation and symmetry) were imposed rather than learned. Was this necessary? Can the model make (testable) predictions about connectomics data?

Thank you for these two questions. The first question is whether symmetries would arise naturally if not imposed. In our optimization, we imposed rotational and mirror symmetries on the excitatory and inhibitory weights, and also aligned the upward directions when there were multiple units in the models. At the same time, we also chose our stimuli to be isotropic: all stimulus positions were distributed at random across visual angles. This is a reasonable null distribution in the absence of information about the true distribution of looming stimuli. But this isotropy means our basal expectation is that because there are no left-right or up-down asymmetries in the stimuli, we expect the filters to be largely symmetric, especially in the model with only one unit *M* = 1.

There is one large advantage to training our models with imposed symmetries: it reduces the total number of parameters in the model by almost by eight-fold. In terms of training examples per parameter, one may also think of this as effectively decreasing the data required for training by eight-fold. This computational efficiency justified our use of the symmetries in the paper.

To directly answer this question, we retrained our *M* = 1 model, this time without imposing the rotation and mirror symmetries (Figure 5—figure supplement 3 in the revised manuscript). As this figure shows, the trained weights possess roughly the rotational and mirror symmetries noted above. To see this more clearly, we quantified the degree of symmetry in these learned filters. To do this, we averaged the filters of the four directions (rotated accordingly to be aligned) and averaged the two halves of each filter. In this way, we created a symmetrized weighting, which we compared to the original unsymmetrized weights by calculating the cosine distance between them. The distribution of the cosine distances (Figure 5—figure supplement 3) shows that most of the trained weights are very close to the symmetrized one. In addition, when we trained the model using a data set eightfold larger than before, the trained weights become more symmetric (Figure 5—figure supplement 3). These results suggest that the isotropic training will naturally lead to the symmetrized weights, and imposing these symmetries in advance will make the training more efficient.

When we align multiple units on a sphere, with upward filters all pointing north, we break the isotropy of the model. In this case, it is possible that solutions without imposed filter symmetry would have asymmetric filters. However, those asymmetries would depend on the details of how we align and position our units. Those unit positions and their alignment, especially near the poles, is not well constrained by data, so we would be extremely reluctant to interpret any asymmetries that arose from our choices about how to distribute and align units. Thus, we do not believe it would be profitable to interpret any such asymmetries or try to tie them to the connectome.

A further important difficulty in comparing our results to connectomic data is that the hemibrain dataset from Janelia regrettably cut off most of the lobula plate, so that there is limited ultra-structure information about LPLC2 dendrites in that brain region. In the future, more ultra-structure data on LPLC2 would be useful both in constraining models of the sort we use here and in testing model predictions.

2. Types of solutions. The text and results needs to explore all three types of solution (inward, outward and unstructured) in more detail. It is currently difficult to understand why the inward and unstructured solutions are essentially dropped part way through.

Thank you for this suggestion. Although we never closely analyzed the unstructured data, in our original submission we did analyze both outward and inward solutions until the very last figure in the main text. In that last figure, we only showed the comparison of the outward solutions with the experimental data. In the revised manuscript, we now provide a supplementary figure for the last figure (Figure 10-supplemental figure 1) that shows the comparison of the response curves of the inward model with the experimental data. Thus, we now follow the inward and outward solutions through to the end of the paper.

In the revised manuscript, we have also addressed the unstructured solution. In the original manuscript, we created some confusion by calling these ’unstructured solutions’, when a more accurate term would have been ’zero solutions’. The spatial weights of the unstructured solutions are exactly zero or very close to zero, and thus, a rather uninteresting solution. In the revised manuscript, we have updated Figure 5 to include the zero solutions labeled in black boxes (Figure 5—figure supplement 1). We also now more clearly describe the unstructured solutions in the first paragraph of the section ’Optimization finds two distinct solutions to the loom-inference problem’. In the new manuscript, we also refer to these solutions as ’zero solutions’ to reduce the confusion brought about by ’unstructured’.

3. More challenging tests of the model. Can you add distracting optic flow to the current stimulus set and/or use more naturalistic stimuli? This could help reduce the number of viable solutions.

This is an interesting suggestion. When we were designing the stimuli used for training, it was not clear to us what naturalistic looming stimuli should look like. It was also not clear what the statistics of the stimuli are and how they should be distributed. Thus, it is not easy to systematically engineer stimuli that are close to the ones that a fly could experience in reality.

However, following this suggestion, we have engineered a new set of stimuli, where for the hit, miss, and retreat cases, we added a rotational background. In these new stimuli, the object of interest, i.e., the one that is moving in the depth direction, also rotates with the background. This mimics the effect of self rotation of the fly while observing a looming or retreating object. In a new training set, we replaced half of the hit cases, half of the miss cases and half of the retreat cases in the original data set with these rotational ones. When optimizing with this more challenging dataset, the outward and inward solutions both continue to exist, and we did observe an expected decrease of the model performance, with a larger decrease for the inward solutions. We present the optimization to this more challenging training set in Figure9—figure supplement 3 and discuss this in the Results section ’Large populations of units improve performance’.

There are two important points to make about interpreting these results. First, we are not aware of any experiments measuring LPLC2 responses to loom with additional background flow fields. Thus, it is not clear that LPLC2 can even perform this task, which makes it difficult to connect these results to data. We are also not aware of data on insect performance to looming stimuli of this more difficult type, so it is also difficult to relate to a true task of an insect. The second point for naturalistic stimuli is that, as mentioned in the Discussion section of the manuscript, the input to our model is not the optical signal itself, but rather the flow field that is calculated by a motion estimator. We have used the simple Reichardt correlator to estimate the motion signals. A more thorough future investigation might examine a more realistic motion detection model that can deal with more complicated, naturalistic visual stimuli, including textures and natural scene statistics, which could add new signals to the inputs to the model.

4. Inhibitory component of the model. Inhibition is assumed to have specific properties (e.g. rectification) – and it is not clear if these are essential. Further, it is absent in some solutions. Are the properties of inhibition (when present) consistent with the broad LPi receptive fields?

This comment and question led to our major change to the linear receptive field model. It is true that it was not clear if the rectified inhibition was necessary, and in fact it was not. As we discussed in our revision summary, we have replaced the model that has a rectified inhibitory component, in our original submission, with a simpler model that has a linear receptive field (Figure 4A). This new model performs almost identically to the previous model, in terms of both AUC performance and replication of the neural data (Figure 9 and supplements, Figure 10 and supplements). The primary difference we observed using the linear receptive field model is that the ratio between the numbers of the outward and inward solutions does not increase as the number of units increases (Figure9—figure supplement 2), and there are fewer outward solutions than inward ones. The nonlinearity of the inhibitory components plays an important role in selecting the outward solutions over the inward ones. Interestingly, if we replace the rectified linear unit (ReLU) with an exponential linear unit (ELU), which has a negative slope below the threshold, for a small number of units, all solutions are inward. But when the number of units increases and become larger than 16, the outward solutions emerge more often from the training. The ratio in this case remains below 1. Combined, these results indicate that the form and position of the nonlinearity in the circuit play a role in selecting between different optimized solutions. This suggests that further studies of the nonlinearities may lead to additional insight into how a population of LPLC2s encodes looming stimuli.

This question also mentions the fact that in some outward solutions, the inhibitory components are zero. In the revised manuscript, for the simpler linear receptive field model, the inhibitory negative component exists in all outward solutions. While it is interesting to examine the family of solutions from the rectified inhibition model, they are no longer the focus of the paper and not central to its claims. In the interest of length, we have not added to our paper by analyzing this specific type of outward solution.

The second question above is whether the inhibitory fields are consistent with broad LPi receptive fields. Here, the new linear model shows that the negative regions (stronger inhibition) are generally broader than the positive regions (stronger excitation) when the number of units is large. In the rectified inhibition model, where excitatory and inhibitory components are dissociable, the inhibitory weights of outward solutions spanned most of the 60-degree receptive field. For both models, the inhibition in outward solutions extended out far enough that responses were inhibited by motion far from looming centers (Figure 10 panels E, F and Figure 10—figure supplement 2 E, F), as in the data from LPLC2.

It is a bit difficult to compare these results directly to LPi data. LPis extend over regions that are similar in size to LPLC2 dendrites (Klapoetke et al., Nature, 2017, Figure 5K and Extended Data Figure 9). Moreover, it is not clear how much LPi cells integrate over space, or whether can have more localized input-output signals, as in neurons like CT1. Overall, for models with large number of units, the inhibition is broader than the excitation and seems consistent with broad (averaged) inputs from LPi neurons.

5. Comparison of model with neural data. A stronger rationale is needed for why two of the many outward models are selected for comparison with neural data (and why comparisons are not made for the inward or unstructured models). It is also important to quantify the similarity of the models with neural data.

Thank you for these suggestions. One advantage of the new linear receptive field model is that the variability in the solutions is mostly eliminated. In the revised manuscript, we now show the outward solution comparison with the data in the main figure (Figure 10) and the inward solution comparison in Figure10—figure supplement 1. We now include the rectified inhibition model comparison in a supplementary figure (Figure 10—figure supplement 2).

In the initial submission, in which there was a distribution of solutions, it might have been useful to quantify the relative fits of the different outward solutions. But with the linear receptive field model, this within-model quantification does not seem warranted because there is no distribution of trained filters.

Our goal in the comparison between the model and the data is to see how the two compare *qualitatively*, rather than quantitatively. For a quantification of the comparison to be interpretable, one would have to account for calcium indicator dynamics, which we have not included in our model. The important points in comparing the model to the data are: (1) the model responds strongly and selectively to loom signals rather than other non-looming signals; (2) the model qualitatively reproduces LPLC2 responses to various expanding bar stmuli; (3) the model shows periphery inhibition as observed in experiments; and (4) the model shows similar size tuning properties to LPLC2 neurons. We outline these qualitative similarities in the text analyzing the data in Figure 10. In general, we are strong proponents of quantifying similarities and differences, but in this case a quantification of these qualitative results does not seem as though it would provide additional insight.

Reviewer #1 (Recommendations for the authors):Line 26-27: It would be helpful to make a somewhat more general statement about the power of the approach that you take here.

We have added a more general statement here, and expanded later in the introduction on how this approach relates to others.

Figure 3 is the first figure referred to, so moving it up to Figure 1 would make reading easier.

We want to keep the anatomy as the first figure, and so we removed the reference to Figure 3 in the first paragraph of the introduction.

Line 79: clarify here you mean object motion, not motion of one of the edges.

We rewrote the sentence to make it more clear that it is object motion.

Line 94-95: the relationship between timing and size-to-speed ratio is likely hard for most readers to make sense of here – suggest deleting.

Removed.

Lines 150-151: suggest clarifying that excitation and inhibition in the model are not constrained to have opposite spatial dependencies as depicted in the Figure 4.

We have added some sentences in both the main text (model section in the results) and the model figure caption to clarify this.

Line 170: suggest describing the loss function in a sentence in the Results.

Did as suggested in the last paragraph of the Results section ’An anatomically-constrained mathematical model’.

Lines 174-176: It would be helpful to connect the outward and inward model terminology more clearly to the flow fields in Figure 3 here. I think this is just a matter of highlighting which elements of the grid in Figure 3 are relevant for each model.

In the revised manuscript, these connections are made in the last two paragraphs of Results section ’Optimization finds two distinct solutions to the loom-inference problem’.

Lines 177-178: describe performance measures here qualitatively.

We have added this.

Lines 206-209: the reason for the difference in baseline activity is not clear – and it requires a lot of effort to extract that from the methods. Can you give more intuition here in the results?

Thank you for highlighting this. Yes, it does require the details of the model to think through this. The baseline activity of the inward solutions does not have to be positive, but it just happens to be. We have added some comments on this in the section ’Outward and inward filters are selective to signals in different ranges of angles’.

Lines 336-340: this is helpful, and some of it could come up earlier in the Results. More generally, it would be helpful to be clearer (especially in results) how much of the encoding of angular size is a property of expansion of the stimulus, and how much of how the computation is implemented.

These comments have been moved to earlier the Results section ’Activation patterns of computational solutions resemble biological responses’. With these comments, we want to provide an intuitive explanation of why the LPLC2 neurons and our models are angular size encoder, but it is not straightforward to quantify the contributions of the two aspects to the angular size tuning.

Reviewer #2 (Recommendations for the authors):– The manuscript is a bit difficult to understand. The authors may want to improve their explanations and figures to make them more accessible. For example, in Figure 7B, I can barely see the responses and don't see any grey lines. Perhaps showing only a subset of responses would make the figure clearer -- less is more.

We have made the lines thicker and panels larger to make the figures clearer.

– The usage of the term "ballistic" in the introduction is confusing. In many contexts, "ballistic" suggests free-falling motion; in this paper, the authors are referring to the distinction between ballistic and diffusive motion. To avoid confusion, I would suggest not using the term ballistic at all; instead, "straight line" or "linear" is just as expressive.

We agree this was inappropriate. We now use the suggested term ”straight line motion”.

– The first figure that is cited in the text is Figure 3. I suggest reorganizing either the text or the figures so that the first figure that is cited is Figure 1.

We have deleted the reference to the Figure 3 in the first paragraph of the introduction.

– Figure 5, panel D: why are there two magenta curves?

In the initial submission, there were more than one example. In the new linear receptive field model, the curves are on top of each other, so there is only one curve apparent. We now state in figure captions when curves lie on top of one another.

– I would also suggest a careful reading to screen for typos -- I found a dozen or so, from misspelled words to mismatched parentheses.

We have read carefully through the manuscript and attempted to find and correct all typos.

Reviewer #3 (Recommendations for the authors):1. Suggestions for improved or additional experiments, data or analyses:a. The authors should provide their criteria for selecting a particular solution to compare to neural data.

Please see Essential Revisions 5. The new linear RF model means that we no longer deal with this distribution of solutions for the main model we study, and selection is not required. Moreover, we now show all three different subtypes of outward solutions for the rectified inhibition model in Figure10—figure supplement 2, 3, 4.

b. The authors should evaluate how well their solutions predict neural data.

Please see Essential Revisions 5. We believe that the qualitative evaluation of the model with data is extremely informative, and without a family of solutions, we are not sure of the goal of a more formal, quantitative comparison between model and data.

c. The authors need to mention that certain outward solutions have no inhibitory component (see Figure 5C, Figure 6 supplement 2). It needs to be discussed in the text and it would be very interesting to see how well these solutions recreate actual data.

The inhibition-absent outward solutions only exist in the rectified inhibition models, but not in the new linear receptive field models. The outward solution without inhibitory component will respond strongly to the moving gratings in Figure 10—figure supplement 4B (which is different from experimental observations), and it cannot show the periphery inhibition in Figure 10—figure supplement 4E and F. We now mention this as among the family outward solutions in the rectified inhibition model, and point out its short-comings.

d. It would be helpful for the authors to provide an example of an "unstructured" solution and an evaluation of its performance, even if it is included as a supplemental figure.

This is now provided in the Figure 5 supplemental figure 1, shown as zero solutions. Please see Essential Revisions 2.

2. Recommendations for improving writing and presentationa. Lines 89-90 – this can be better supported by adding the criteria/evaluation mentioned above.

Thank you for this suggestion. We have added more detail about the evaluations of the models in the Results section ’Optimization finds two distinct solutions to the loom-inference problem’.

b. Methods (~ line 483) – How is the HRC model using T5 (off) and T4 (on) motion input?

The HRC model we use does not distinguish between light and dark edges. Using it as the input is most similar to having both T4 and T5 input (which is also why HS cell activity can often be well-approximated by an HRC).

c. Lines 492-502 – What was the frame rate (timestep) for both training and testing stimuli?

We have added this information in the methods: the time step for the stimuli is also 0.01 second.

d. Figures – Please increase the size when there is white space available. Make sure the pink and green color scheme for the two solution sets are very obvious.

Increased the sizes of some panels.

e. Figure 1 caption – approximately half of the 200 LPLC2 are directly synaptic to the GF.

We are uncertain where this information comes from. In the Ache et al., paper (Current Biology, 2019), they reported 108 LPLC2 neurons projecting to the GF in the right hemisphere of an adult *Drosophila*. So, in total, there should be about 200 LPLC2 neurons directly projecting to the two GFs. In the hemibrain dataset, there are 68 annotated LPLC2-R neurons and all 68 LPLC2-R neurons are listed a presynaptic to the right giant fiber in a neuprint query. When not restricted to the ’-R’ suffix, one finds a similarly large fraction of LPLC2 neurons presynaptic to the giant fiber. Unless we are mistaken, it appears that most LPLC2 neurons synapse onto the GF. In the Figure 1 caption and introduction, we changed GF to GFs to indicate that these 200 LPLC2 project to two GFs, respectively. If we have missed an important measurement of this connectivity, we would be happy to correct this description if the reviewer could provide the reference.

f. Figure 5 – is cross entropy loss the same as what is referred to as the loss function (equation 6) in the methods? If so, keep consistent. If not, please explain.

Yes, they are the same. We have changed the l.h.s of Equation 6 from loss to cross entropy loss.

g. Figure 8D, it is difficult to see the boxplots.

In the revised manuscript, we have made the boxes larger and hopefully easier to see.

h. Figure 10 I-L, it is difficult at first glance to realize what is neural data vs model output. Maybe label the rows instead?

We have labeled the rows as suggested.

i. Supplemental Figure 1. Add a schematic for the HRC model for readers who may not be familiar with it.

Added as suggested.